# CENTRALITY-GUIDED PRE-TRAINING FOR GRAPH

**Bin Liang**[1,3,5*]**, Shiwei Chen**[2,3*]**, Lin Gui**[4†]**, Hui Wang**[2]**, Yue Yu**[2]**,**
**Ruifeng Xu**[2,3†]**, Kam-Fai Wong**[1,5†]
[1]The Chinese University of Hong Kong    [2]Peng Cheng Laboratory
[3]Harbin Institute of Technology (Shenzhen)    [4]King's College London
[5]Ministry of Education Key Laboratory of High Confidence Software Technologies, CUHK
`bin.liang@cuhk.edu.hk; {shwchen, wangh06, yuy}@pcl.ac.cn;`
`Lin.1.gui@kcl.ac.uk; xuruifeng@hit.edu.cn; kfwong@se.cuhk.edu.hk`

## ABSTRACT

Self-supervised learning (SSL) has shown great potential in learning generalizable representations for graph-structured data. However, existing SSL-based graph pre-training methods largely focus on improving graph representations by learning the structure information based on disturbing or reconstructing graphs, which ignores an important issue: the importance of different nodes in the graph structure may vary. To fill this gap, we propose a Centrality-guided Graph Pre-training (CenPre) framework to integrate the distinct importance of nodes in graph structure into the corresponding representations of nodes based on the centrality in graph theory. In this way, the different roles played by different nodes can be effectively leveraged when learning graph structure. The proposed CenPre contains three modules for node representation pre-training and alignment. The first is a node-level importance learning module, fusing the fine-grained node importance into node representation based on degree centrality, allowing the aggregation of node representations with equal/similar importance. The second one, the graph-level importance learning module, characterizes the importance between all nodes in the graph based on eigenvector centrality, enabling the exploitation of graph-level structure similarities/differences when learning node representation. Finally, a representation alignment module aligns the pre-trained node representation using the original one, permitting graph representations to learn structural information without losing their original semantic information, thereby leading to better graph representations. Extensive experiments on a series of real-world datasets demonstrate that the proposed CenPre outperforms the state-of-the-art baselines in the tasks of node classification, link prediction, and graph classification[1].

## 1 INTRODUCTION

Graph neural networks (GNNs) aim to model the structural information of the graph by neighborhood aggregation schemes, becoming increasingly popular in graph representation learning for graph-structured data (Zhu et al., 2021a), such as knowledge graphs (Baek et al., 2020), social networks (Fan et al., 2019), point clouds (Shi & Rajkumar, 2020), and chemical analysis (De Cao & Kipf, 2018). Graph representation learning can produce low-dimensional vector representations for graph-structured data in many applications, including node/graph classification, link prediction, and graph generation (You et al., 2020; Wang et al., 2020; Zhang & Chen, 2018; Hou et al., 2019).

Recently, self-supervised learning on graph-structured data has shown great potential in learning generalizable, transferable, and robust representations due to the advantage of learning graph structural information without annotated data (Hu* et al., 2020; You et al., 2020). Among many, graph predictive learning and graph contrastive learning (GCL) have become two main paradigms in learning graph representation, where mutual information between graph representations can be learned by leveraging the similarity/difference between augmented views (Hu* et al., 2020; Kim et al., 2022; You et al.,

---

[*]Equal contribution.
[†]Corresponding authors.
[1]The source code of this work is released at `https://github.com/HITSZ-HLT/CenPre`.

2020; 2021; Xia et al., 2022b). Furthermore, Jin et al. (2020) and Hu et al. (2019b) explore centrality in learning graph structures in pre-training graphs. Despite promising progress made by existing graph representation learning methods, they largely ignore the importance of different nodes in the graph structure may vary when learning graph structure. This deficiency may result in the inability to exploit important characteristic information in the graph structure when learning graph representations.

To alleviate this issue, in this paper, we propose a novel graph pre-training framework to produce better graph representation that leverages graph structure in learning graph representations by exploiting the distinct importance of different nodes in graph structure based on the notion of graph centrality, dubbed Centrality-guided Graph Pre-training (CenPre). There are three modules in the proposed CenPre: 1) **node-level importance learning** module, which aims to enhance the node representations with the node importance in graph structure based on the property of degree centrality; 2) **graph-level importance learning** module, which is designed to leverage the global importance of a node based on the property of eigenvector centrality when learning node representation; and 3) **graph representation alignment** module, which aligns the importance-fused node representation with the original one.

**Node-level importance learning**. Degree centrality defines the importance of a node based on the degree of that node. The higher the degree, the more crucial a node becomes in the graph (Zaki & Meira, 2014). Based on this property, we argue that the degree of a node can represent its importance in a graph, which plays a crucial role in its representation, so its representation can be refined by degree. Therefore, we explore predictive learning to map each node representation into the corresponding degree. This approach enables the model to capture statistical regularities between representations and graph structures, allowing it to assess node importance effectively. As a result, nodes with similar degrees receive comparable representations, while distinctions are maintained for nodes of varying significance.

**Graph-level importance learning**. Eigenvector centrality, or eigencentrality, measures a node's influence within a connected network by accounting for direct and indirect connections (Zaki & Meira, 2014; Bonacich, 2007). Such a measure provides a comprehensive understanding of a node's role in the global graph structure. Building on this, we propose identifying the most influential neighbors of each node to obtain a global structural perspective, which can guide the refinement of node representations. To achieve this, we introduce a Contrastive Representation-Structure Pre-Training (CReSP) strategy that aligns node representations with the graph-level importance of nodes. Specifically, matrix decomposition is used to extract significant eigenvalues and eigenvectors, facilitating the identification of key neighbors in large, sparse graphs. Using cross-attention, the graph representation guides the structure matrix in determining node importance relative to the current node. Inspired by CLIP (Radford et al., 2021), CReSP refines this alignment by matching node representations with their structural patterns, maximizing the similarity between nodes with shared important neighbors and distinguishing node representations with different global importance, thereby enhancing node representations by incorporating graph-level node importance information.

**Graph representation alignment**. After the graph structure pre-training, a graph representation alignment module is devised to align the pre-trained structure-fused graph representation using the original representation, allowing the graph representation to retain the original semantic information while learning the graph structure information.

Our main contributions are as follows:

- We are the first to explore node importance in learning graph structure and align the graph representation with graph structure in the graph pre-training process, aiming to produce better graph representations for downstream tasks.

- Based on the notion of centrality, a novel Centrality-guided Graph Pre-training (CenPre) framework is proposed to learn importance-fused graph representation from both local and global perspectives.

- We conduct a series of experiments on real-world graph-structured benchmark datasets to evaluate the effectiveness of our CenPre in learning graph representation. Experimental results show that our CenPre significantly outperforms baselines in the tasks of node classification, link prediction, and graph classification.

## 2 RELATED WORK

### 2.1 GRAPH NEURAL NETWORKS

Graph Neural Networks (GNNs) capture node dependencies through graph topology. The Graph Convolutional Network (GCN) (Kipf & Welling, 2017) aggregates local neighborhood features, excelling in node classification but struggling with global structures. Graph Attention Network (GAT) (Velickovic et al., 2018) improves feature aggregation via attention mechanisms, enhancing performance in heterogeneous graphs but increasing computational demands. Graph Isomorphism Network (GIN) (Xu et al., 2019) improves structure distinction but risks overfitting on small datasets, while GraphSAGE (Hamilton et al., 2017) scales well for large graphs using neighborhood sampling, though it may lose information in dense networks. Graph Transformers (Dwivedi & Bresson, 2020) capture long-range dependencies with self-attention, but are computationally intensive for large-scale graphs. Some other previous works extend GNNs by leveraging centrality to enhance structural understanding. For instance, Maurya et al., 2019 proposes a GNN framework for approximating betweenness centrality by leveraging constrained message passing and ranking loss to learn structural node importance efficiently. (Avelar et al., 2018) introduces a multitask GNN-based learning framework to approximate multiple centrality measures, enabling shared representations and accurate structural feature predictions.

### 2.2 SELF-SUPERVISED LEARNING ON GRAPHS

Self-supervised learning has achieved promising performance in graph pre-training. Early works like DeepWalk (Perozzi et al., 2014) and node2vec (Grover & Leskovec, 2016) use random walks to capture local structures, while contrastive learning methods such as DGI (Veličković et al., 2019), InfoGraph (Sun et al., 2020), and GraphCL (You et al., 2020) maximize agreement between augmented views. MVGRL (Hassani & Khasahmadi, 2020) contrasts different graph views using diffusion, but these methods face challenges in selecting augmentations and handling negative samples. GGD (Zheng et al., 2022) eliminates contrastive similarity computation by discriminating augmented positive and negative node groups through a binary classification task. BGRL (Thakoor et al., 2021), T-BGRL (Shiao et al., 2022), and CCA-SSG (Zhang et al., 2021) align representations without negative samples, offering more efficient alternatives. Graph autoencoders (GAE) (García-Durán & Niepert, 2017), VGAE (Kipf & Welling, 2016), and other variants reconstruct graph structure but often underperform in classification. GraphMAE (Hou et al., 2022) improves performance using masked feature reconstruction, bypassing augmentations, while GPT-GNN (Hu et al., 2020) introduces autoregressive graph generation for pre-training. Along the line of work on multi-task learning, GBT (Bielak et al., 2021) proposes to incorporate multi-task objectives, including node proximity and subgraph features, to capture intrinsic graph properties without relying on augmentation. PARETOGNN (Ju et al., 2022) enhances task generalization by reconciling multiple pretext tasks through a multiple-gradient descent algorithm promoting Pareto optimality. AutoSSL (Jin et al., 2021) leverages the homophily principle to automatically search for optimal combinations of multiple self-supervised tasks, significantly enhancing the performance on downstream tasks such as node clustering and classification. For the works that use centrality, GCA (Zhu et al., 2021b) adaptively retains critical structural features using centrality-guided augmentations to enhance representation learning. Relying on node degrees for its centrality-based learning, Jin et al. (2020) combines tasks such as clustering and node distance prediction to learn robust structural embeddings. Hu et al. (2019b) uses centrality score ranking to guide GNNs in capturing structural features but overlooks the actual value of the centrality score. Unlike previous approaches, we use degree centrality to assess node importance and Eigenvector centrality for global structural information, integrating both local and global properties to guide graph pre-training efficiently. A detailed analysis of our motivation is shown in Appendix F.

## 3 PRELIMINARIES

**Notations** Let $\mathcal{G} = \{\mathcal{V}, \mathcal{E}\}$ represent an undirected graph, where $\mathcal{V} = \{v_1, v_2, ..., v_n\}$ is the set of nodes and $\mathcal{E}$ the set of edges. $X_v \in \mathbb{R}^{N \times d_v}$ and $X_e \in \mathbb{R}^{|\mathcal{E}| \times d_e}$ denote the node and edge feature matrices, respectively. The representation of a node $v_i$ is $h_i$, and the graph-level representation is $H_{\mathcal{G}} = \{h_1, h_2, ..., h_n\}$.

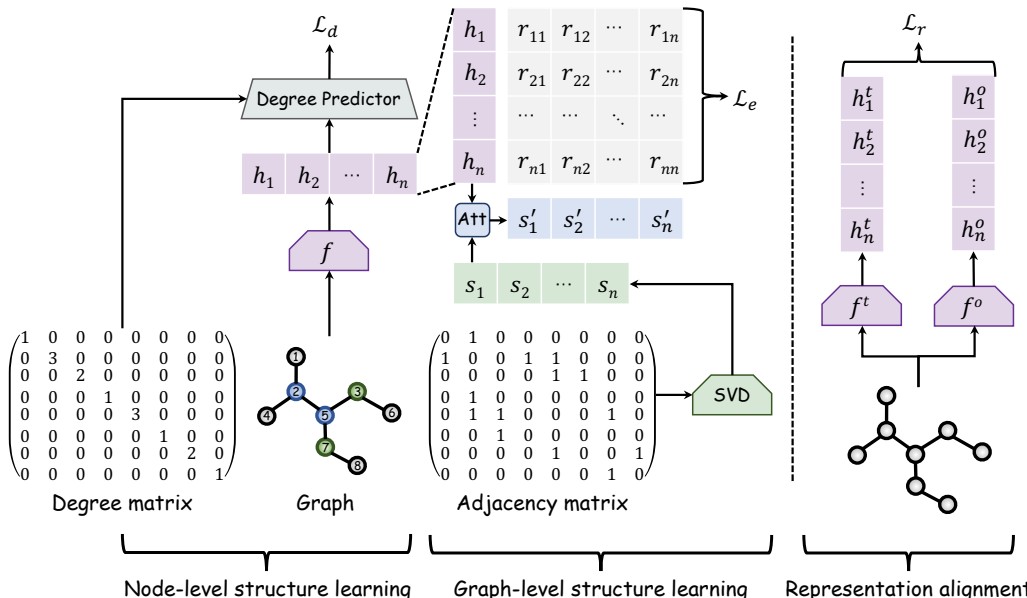

Figure 1: Illustration of our CenPre framework. $f$ is the GNN encoder that is used to generate node representation, $f^t$ is the pre-trained GNN encoder produced by our CenPre framework, and $f^o$ is the original GNN encoder without pre-training."Att" represents cross-attention. $h$, $s$, and $r$ are embeddings, where $r_{ii} = h_i \cdot s_i'$.

**Graph Neural Networks**   GNNs update the graph representation $H_{\mathcal{G}}$ by leveraging the graph topology. For example, GCN (Kipf & Welling, 2016) updates node representations by aggregating neighborhood information based on the adjacency matrix $A$. The degree matrix $D$, with $D_{ii}$ representing the degree of node $v_i$, normalizes the influence of neighboring nodes. The propagation rule for GCN is:

$$H^{(K)} = \sigma\left(\tilde{D}^{-\frac{1}{2}}\tilde{A}\tilde{D}^{-\frac{1}{2}}H^{(K-1)}W^{(K)}\right) \tag{1}$$

where $\tilde{A} = A + I$ includes self-loops, $\tilde{D}$ is its degree matrix, $W^{(K)}$ is the trainable weight matrix, and $\sigma(\cdot)$ is a non-linear activation function.

## 4  METHOD

The motivation for CenPre is based on node2vec (Grover & Leskovec, 2016), which suggests that nodes with similar roles have similar embeddings. Additionally, as highlighted in (McNulty, 2022), influential nodes in graph layouts tend to be positioned centrally due to their stronger connectivity. Building on the relationship between node importance and centrality, we propose a novel Centrality-guided Graph Pre-training (CenPre) framework to learn node representations. Unlike node2vec (Grover & Leskovec, 2016), our goal is to enhance node embeddings by aggregating or differentiating them based on centrality, rather than just role similarity. The CenPre architecture is shown in Figure 1.

To achieve this, we propose integrating node importance into representation learning from two perspectives: 1) **node-level importance learning**, which enhances node representations based on their individual importance in the graph; and 2) **graph-level importance learning**, which improves node representations by considering the importance of related nodes. We introduce a **graph representation alignment** module to align the centrality-guided node representations with the original ones, preserving semantic information while incorporating node importance. In this study, we use degree centrality and eigenvector centrality as local and global measures of node importance, respectively, to guide CenPre's pre-training for more accurate, structure-aware node embeddings. More discussion on other centrality measures can be found in Appendix A.

## 4.1 Node-level Importance Learning

In this section, we introduce the node-level importance learning module of our CenPre framework in detail. Based on the Definition 1 from Degree Centrality, we propose node-level importance learning, aiming to identify the degrees of nodes to distinguish the importance of nodes, so as to guide the refinement of the node representation according to the degree of the node.

**Definition 1.** *Degree Centrality defines the importance of a node based on the degree of that node. The higher the degree, the more crucial it becomes in the graph.*

From Eq.1, it is evident that GNNs like GCNs use the adjacency matrix to aggregate information from neighboring nodes in an equal and uniform manner, assuming that all sources of information contribute equally. However, GCNs focus solely on the connections between nodes when aggregating neighbor information and do not account for the specific roles or characteristics of nodes. In this study, we address this limitation by incorporating node importance into the learning process, allowing for more accurate representations. To be specific, for each graph, we obtain the degree $d_i$ of each node $v_i$ by summing the values in its corresponding row of the adjacency matrix $\mathcal{A}$, which represents the connections between nodes:

$$\mathcal{Y}_i^d = d_i = \sum_j \mathcal{A}_{ij} \tag{2}$$

where $\mathcal{A}_{ij}$ represents the entry in the adjacency matrix $\mathcal{A}$ indicating the presence (1) or absence (0) of an edge between nodes $v_i$ and $v_j$. Then, we train a degree predictor $\mathcal{P}_d$ to predict the degree of each node based on the node representations $\{h_1, h_2, ..., h_n\}$. The loss of $\mathcal{P}_d$ is defined as:

$$\mathcal{L}_d = -\frac{1}{n}\sum_{i=1}^{n}\mathcal{Y}_i^d\log\big(\mathcal{P}_d(h_i; \theta_d)\big) \tag{3}$$

where $\mathcal{P}_d(h_i; \theta_d)$ is the predicted distribution of the degree of node $v_i$ and $\mathcal{Y}_i^d$ is the ground-truth distribution. $\theta_d$ represents the trainable parameters of the degree predictor $\mathcal{P}_d$. In this way, the degree information can be integrated to refine the node representation.

## 4.2 Graph-level Importance Learning

Node-level importance learning reflects a local perspective, focusing on the importance derived from the node itself. Solely utilizing node degree as a metric for node-level learning might be misleading, as nodes with similar degrees are not necessarily similar in other respects (e.g., two users in a social network can have the same number of followers). A broader, more global perspective that considers attributes like "the degree distribution of a node's neighbors" is necessary. Such an intuition is captured by the essence of Eigenvector Centrality (Definition 2), which reflects how a node's importance is shaped by the importance of its neighbors. Therefore, beyond learning local node-level importance, we propose to integrate graph-level importance using Eigenvector Centrality.

**Definition 2.** *Eigenvector Centrality defines relationships with high-scoring nodes have more contribution to the score of a node than connections to nodes with low eigenvector centrality scores.*

As described previously and by definition, degree centrality provides insight into a node's immediate environment, while eigenvector centrality reflects its importance relative to the entire network. Together, they offer a more comprehensive picture of a node's role within the graph, which is naturally divided into four categories. More details are shown in Appendix A.

To compute Eigenvector Centrality, we first obtain $\lambda_{\max}$, which is the maximum absolute eigenvalue of the adjacency matrix $\mathcal{A}$, and solve for the eigenvector $\vec{v}$:

$$\lambda_{\max}\vec{v} = \mathcal{A}\vec{v} \tag{4}$$

This is an example of the Eigen-Decomposition, where we decompose $\mathcal{A}$ into a set of eigenvectors and eigenvalues. This classic technique is powerful but may not always be numerically stable for ill-conditioned matrices[2], which can arise in large, sparse, or noisy networks. Moreover, it is also

---

[2]An ill-conditioned matrix typically has a high condition number, which is the ratio of the largest to the smallest singular value. This makes numerical operations such as Inversion or Eigen-Decomposition highly sensitive to small perturbations in the data.

computationally expensive for large matrices. To address this, we turn to (Truncated) Singular Value Decomposition (ED and SVD are equal if the $\mathcal{A}^{\mathsf{T}} = \mathcal{A} \wedge \mathcal{A} \succeq 0$, which is usually the case for $\mathcal{A}$ from undirected graph, see Appendix B for a proof). It retains only the top $k$ singular values and vectors[3], reducing computational cost while preserving the most important structural information:

$$\mathcal{A} = U\Sigma V^{\mathsf{T}} \approx U_k \Sigma_k V_k^{\mathsf{T}} \tag{5}$$

where $U$ and $V$ are orthogonal matrices containing the left and right singular vectors, and $\Sigma$ is a diagonal matrix storing the singular values. The left singular vectors in $U_k$ can be used as structural representations for each node. Specifically, each row of the matrix $U_k$ provides a $k$-dimensional embedding that captures the most important structural properties of the graph with respect to its overall connectivity. Denoting this structural representation retrieval process as a function $f_{\text{truncated\_svd}}$ that returns the truncated left singular matrix $U_k$ for a given adjacency matrix $\mathcal{A}$, we can obtain the structural representation $s_i$ for each node $v_i$ as follows:

$$S_{\mathcal{G}} = \{s_1, s_2, \ldots, s_n\} = \{U_{0,*}^k, U_{1,*}^k, \ldots, U_{n,*}^k\} = U_k = f_{\text{truncated\_svd}}(\mathcal{A}) \tag{6}$$

Based on this, the issue of producing importance-fused structure representation for a specific node $v_i$ evolved into how to determine the importance of each node in a graph for $v_i$. In our case, we treat the structure representation $S_{\mathcal{G}}$ and graph representation $H_{\mathcal{G}}$ as two modalities–one encoding the structural properties and the other capturing the feature-based characteristics of the graph–and use graph representation $H_{\mathcal{G}}$ to cross-attend over structural representation. By aligning the two modalities, we produce the importance-fused graph-level structural representation $S_{\mathcal{G}}'$:

$$S_{\mathcal{G}}' = \{s_1', s_2'..., s_n'\} = \text{CrossAtt}(H_{\mathcal{G}}; S_{\mathcal{G}}) = \text{Softmax}\big((W_q H_{\mathcal{G}})(W_k S_{\mathcal{G}})\big)(W_v S_{\mathcal{G}}) \tag{7}$$

where $W_q$, $W_k$, and $W_v$ are the weight matrices of query, key, and value in the cross-attention mechanism, respectively. After obtaining the importance-fused structure representation $S_{\mathcal{G}}'$, inspired by CLIP (Radford et al., 2021) aligning text and image modalities, we propose a Contrastive Representation-Structure Pre-Training (CReSP) model $\mathcal{P}_e$ to train the node representation based on the graph-level importance-fused structure representation. This contrastive learning objective is designed to capture both local node features and global structural roles by aligning these two complementary modalities in the same embedding space. The loss of $\mathcal{P}_e$ is defined as:

$$\mathcal{L}_e = -\frac{1}{n}\sum_{i=1}^{n}\frac{1}{2}\Big(\mathcal{Y}_i^e \log\big(\mathcal{P}_e(r_i; \theta_e)\big) + \mathcal{Y}_i^e \log\big(\mathcal{P}_e(r^{\mathsf{T}}_i; \theta_e)\big)\Big), \quad r_i = [r_{i1}, r_{i2}, ..., r_{in}] \tag{8}$$

where $\mathcal{Y}^e$ represents the index label towards the importance-fused structure matrix. $r_{ij} = h_i s_j'$ represents the computation of similarity between $h_i$ and $s_j'$. $\theta_e$ represents the trainable parameters of the CReSP model $\mathcal{P}_e$.

## 4.3 GRAPH REPRESENTATION ALIGNMENT

Based on the pre-trained GNN encoder $f^t$ learned by our CenPre, we use the $L_2$-norm to align the structure-fused node representation with the original one, aiming to prevent the loss of the original semantic information while learning the graph structure information[4]. The loss of graph representation alignment module is defined as:

$$\mathcal{L}_r = \frac{1}{n}\sum_{i=1}^{n}\big|\big|\big(f^t(v_i), f^o(v_i)\big)\big|\big|_2 = \frac{1}{n}\sum_{i=1}^{n}\sqrt{\sum_{j=1}^{d_v}\big(f^t(v_{ij}) - f^o(v_{ij})\big)^2} \tag{9}$$

where $f^t(v_i)$ and $f^o(v_i)$ represent the representation of node $v_i$ produced by the pre-trained GNN encoder $f^t$ and the original GNN encoder $f^o$.

---

[3]Conventionally, we set $k$ to explain $\geq 0.95$ variance.

[4]In the preliminary experiments, we also tried other loss functions, such as KL-divergence, $L_1$-norm, Cosine Distance, etc. We found that the performance of $L_2$-norm was more stable, so we use $L_2$-norm in our method.

## 4.4 OVERALL LEARNING OBJECTIVE

The overall learning objective of our CenPre is to train the framework by jointly minimizing the three losses derived from Node-level Importance Learning, Graph-level Importance Learning, and Graph Representation Alignment. The overall loss $\mathcal{L}$ is defined as:

$$\mathcal{L} = \lambda_1 \mathcal{L}_d + \lambda_2 \mathcal{L}_e + \lambda_3 \mathcal{L}_r \tag{10}$$

where hyperparameters $\lambda_1$, $\lambda_2$ and $\lambda_3$ are scaling weights to balance the losses. In this way, our proposed CenPre can effectively integrate structural information into node representations to produce structure-fused node representation by learning important information from graph structures. Meanwhile, the graph representation alignment that aligns the structure-fused representation using the original node representation can also prevent the loss of original semantic information due to excessive learning of structural information.

## 5 EXPERIMENTS

In this section, we first evaluate the performance of our CenPre compared with existing state-of-the-art (SOTA) competitors in the tasks of node classification, link prediction, and graph classification. Then, we present a deep analysis of our CenPre to show its effectiveness in learning graph representation.

## 5.1 DATASETS AND EXPERIMENTAL SETTINGS

**Dataset** We evaluate the effectiveness of our proposed CenPre framework through the tasks of node classification, link prediction, and graph classification on 13 widely used benchmark datasets. These datasets include the Citation Networks triplet (Kipf & Welling, 2017) (**Cora, Citeseer, Pubmed**), Amazon-Co-Purchase networks (Shchur et al., 2018) (**Computer, Photo**), TUD Benchmark datasets (Morris et al., 2020) (**MUTAG, NCI1, PROTEINS, DD, IMDB-B, RDT-B**), and two

Table 1: Statistics of datasets.

| Datasets | Task | #Graphs | #Nodes | #Edges | #Features | #Classes |
|---|---|---|---|---|---|---|
| Cora | Node&Link | 1 | 2,708 | 10,556 | 1,433 | 7 |
| CiteSeer | Node&Link | 1 | 3,327 | 9,104 | 3,703 | 6 |
| PubMed | Node&Link | 1 | 19,717 | 88,648 | 500 | 3 |
| Computer | Node | 1 | 13,752 | 491,722 | 767 | 10 |
| Photo | Node | 1 | 7,650 | 238,162 | 745 | 8 |
| arXiv | Node | 1 | 169,343 | 2,315,598 | 128 | 40 |
| Collab | Link | 1 | 235,868 | 1,285,465 | 128 | - |
| MUTAG | Graph | 188 | 17.93 | 19.79 | 7 | 2 |
| NCI1 | Graph | 4,110 | 29.87 | 32.30 | 5 | 2 |
| PROTEINS | Graph | 1,113 | 39.06 | 72.82 | 29 | 2 |
| DD | Graph | 1,178 | 284.32 | 715.66 | 7 | 2 |
| IMDB-B | Graph | 1,000 | 19.77 | 96.53 | 10 | 2 |
| RDT-B | Graph | 2,000 | 429.63 | 497.75 | 10 | 2 |

large-scale graphs from the Open Graph Benchmark (Hu et al., 2020) (**ogbn-arXiv, ogbl-Collab**). The statistics of the datasets are shown in Table 1. We can see that these datasets span various domains, with the number of graphs ranging from 1 to 4,110, the average number of nodes ranging from 17.93 to 235,868, and the average number of edges ranging from 19.79 to 2,315,598, demonstrating the diversity and comprehensiveness of the datasets.

**Baselines & Implementation Details** We compare our CenPre with a series of SOTA baseline models, including 1) supervised learning methods: GCN (Kipf & Welling, 2017), GAT (Velickovic et al., 2018), GIN (Xu et al., 2019), and SAGE (Hamilton et al., 2017); 2) graph kernels methods: WL (Shervashidze et al., 2011) and DGK (Yanardag & Vishwanathan, 2015); 3) self-supervised learning methods: node2vec (Grover & Leskovec, 2016), graph2vec (Narayanan et al., 2017), Info-Graph (Sun et al., 2020), GAE (Kipf & Welling, 2016), VGAE (Kipf & Welling, 2016), ARGA (Pan et al., 2018), GraphMAE (Hou et al., 2022), DGI (Veličković et al., 2019), GRACE (Zhu et al., 2020), GCA (Zhu et al., 2021b), BGRL (Thakoor et al., 2021), CCA-SSG (Zhang et al., 2021), GraphCL (You et al., 2020), JOAO (You et al., 2021), InfoGCL (Xu et al., 2021), SimGRACE (Xia et al., 2022a), AutoGCL (Yin et al., 2022), MaskGAE$_e$ (Li et al., 2023), MaskGAE$_p$ (Li et al., 2023), TopoGCL (Chen et al., 2024), Patcher (Ju et al., 2023) and GPA (Zhang et al., 2024). Please refer to Appendix C for detailed information of these baseline models and Appendix D for implementation details, which follows the evaluation protocol established by previous works (Li et al., 2023; Hou et al., 2022).

Table 2: Experimental results of **node classification**. Averaged accuracy±std. (%) over 10 runs are reported. The best and second-best results are highlighted in **red** and **blue**, respectively. A.R. is short for the average rank. The smaller the value of A.R., the higher the ranking of model performance. Results with $\star$ denote the significance tests of our CenPre over the baseline models at $p-$value $< 0.05$.

| | Methods | Cora | CiteSeer | PubMed | Computer | Photo | arXiv | A.R.↓ |
|---|---|---|---|---|---|---|---|---|
| Supervised | GCN | 81.50±0.20 | 70.30±0.40 | 79.00±0.50 | 86.51±0.54 | 92.42±0.22 | 70.40±0.30 | 10.8 |
| | GAT | 83.00±0.70 | 72.50±0.70 | 79.00±0.30 | 86.93±0.29 | 92.56±0.35 | 70.60±0.30 | 8.0 |
| | GAE | 74.90±0.40 | 65.60±0.50 | 74.20±0.30 | 85.10±0.40 | 91.00±0.10 | 63.60±0.50 | 15.0 |
| | VGAE | 76.30±0.20 | 66.80±0.20 | 75.80±0.40 | 85.80±0.30 | 91.50±0.20 | 64.80±0.20 | 14.0 |
| | ARGA | 77.95±0.70 | 64.44±1.19 | 80.44±0.74 | 85.86±0.11 | 91.82±0.08 | 67.34±0.09 | 12.5 |
| Self-supervised | NodeProperty | 81.94±0.00 | 71.60±0.00 | 79.44±0.00 | - | - | - | 10.3 |
| | DGI | 82.30±0.60 | 71.80±0.70 | 76.80±0.60 | 83.95±0.47 | 91.61±0.22 | 65.10±0.40 | 11.8 |
| | GRACE | 81.90±0.40 | 71.20±0.50 | 80.60±0.40 | 86.25±0.25 | 92.15±0.24 | 68.70±0.40 | 10.5 |
| | GCA | 81.80±0.20 | 71.90±0.40 | 81.00±0.30 | 87.85±0.31 | 92.53±0.16 | 68.20±0.20 | 8.8 |
| | BGRL | 82.86±0.49 | 71.41±0.92 | 82.05±0.85 | **90.34**±0.19 | 93.17±0.30 | 71.64±0.12 | 5.7 |
| | CCA-SSG | 83.59±0.73 | 73.36±0.72 | 80.81±0.38 | 88.74±0.28 | 93.14±0.14 | 69.22±0.22 | 6.7 |
| | GraphMAE | 84.20±0.40 | 73.40±0.40 | 81.10±0.40 | 89.51±0.08 | 93.23±0.13 | 71.75±0.17 | 3.8 |
| | Patcher | 84.17±0.54 | 71.65±0.05 | 81.13±0.68 | 89.44±0.79 | 81.23±0.32 | **72.31**±0.22 | 6.7 |
| | MaskGAE$_e$ | 83.77±0.33 | 72.94±0.20 | 82.69±0.31 | 89.44±0.11 | 93.30±0.04 | 70.97±0.29 | 4.5 |
| | MaskGAE$_p$ | **84.30**±0.39 | **73.80**±0.81 | **83.58**±0.45 | 89.54±0.06 | **93.31**±0.13 | 71.16±0.33 | **2.7** |
| | CenPre (ours) | **85.15**±0.49$^\star$ | **76.94**±2.12$^\star$ | **83.91**±0.12 | **91.22**±0.05$^\star$ | **93.96**±0.14 | **72.47**±0.15 | **1.0** |

Table 3: Experimental results of **link prediction**. Average AUC, Average Precision (AP), and Hit@50±std. (%) over 10 runs are reported. Hit@50 measures the proportion of correct links among the top 50 predictions. The best and second-best results are highlighted in **red** and **blue**, respectively.

| | Methods | Cora | | CiteSeer | | PubMed | | COLLAB | A.R.↓ |
|---|---|---|---|---|---|---|---|---|---|
| | | AUC | AP | AUC | AP | AUC | AP | Hit@50 | |
| Supervised | GCN | 86.70±0.20 | 87.55±0.05 | 91.10±0.50 | 91.72±0.43 | 84.66±0.10 | 86.20±0.61 | 47.14±0.01 | 9.4 |
| | GAT | 86.84±0.27 | 88.66±0.08 | 91.20±0.10 | 92.02±0.44 | 84.23±0.10 | 86.62±0.22 | - | 10.1 |
| | GIN | 86.66±0.59 | 87.62±0.58 | 92.62±0.24 | 92.54±0.11 | 84.05±0.32 | 86.17±0.25 | - | 10.4 |
| | SAGE | 86.33±1.06 | 88.81±1.36 | 92.54±0.87 | 92.70±1.02 | 84.98±2.65 | 87.12±2.95 | 54.63±1.12 | 7.3 |
| Latent | node2vec | 78.32±0.74 | 78.91±0.77 | 75.36±1.22 | 76.03±0.11 | 79.98±0.35 | 81.55±0.83 | 57.03±0.52 | 10 |
| | MatrixFactor | 62.25±2.21 | 64.20±1.17 | 61.65±3.80 | 61.99±2.50 | 68.56±12.13 | 68.23±3.13 | 48.96±0.29 | 11.5 |
| Self-supervised | GAE | 91.09±0.01 | 92.83±0.03 | 96.40±0.01 | 96.50±0.02 | 90.52±0.04 | 91.68±0.05 | 47.14±1.45 | 7.1 |
| | VGAE | 91.40±0.01 | 92.60±0.01 | 94.40±0.02 | 94.70±0.02 | 90.80±0.02 | 92.00±0.02 | 45.53±1.87 | 7.8 |
| | ARGA | 92.40±0.00 | 93.23±0.00 | 96.81±0.00 | 97.11±0.00 | 91.94±0.00 | 93.03±0.00 | 28.39±2.51 | 6.3 |
| | GraphMAE | 94.88±0.23 | 93.52±0.51 | 96.24±0.36 | 95.47±0.41 | 94.32±0.40 | 93.54±0.22 | 53.97±0.64 | 5.3 |
| | MaskGAE$_e$ | 96.42±0.17 | 95.91±0.25 | **98.02**±0.22 | **98.18**±0.21 | 98.75±0.04 | 98.66±0.06 | 65.84±0.47 | 2.8 |
| | MaskGAE$_p$ | **96.45**±0.18 | **95.95**±0.21 | 97.87±0.22 | 98.09±0.17 | **98.84**±0.04 | **98.78**±0.05 | **65.98**±0.39 | **2.3** |
| | CenPre (ours) | **97.44**±0.85 | **96.05**±0.65 | **99.02**±0.59$^\star$ | **99.04**±0.37 | **99.82**±0.10$^\star$ | **99.62**±0.10$^\star$ | **66.03**±0.27 | **1.0** |

## 5.2 COMPARISONS WITH BASELINES

**Node Classification** Table 2 shows the results of node classification. Our CenPre overall outperforms the SOTA baseline models, in which our CenPre achieves the best performance on all datasets. This demonstrates the effectiveness of the proposed CenPre in node classification. Further, our CenPre performs best on both the Cora dataset with 2,708 nodes and the arXiv dataset with 169,343 nodes, indicating that our CenPre can be effective on datasets of different types and sizes.

**Link Prediction** We further evaluate the performance of our CenPre in link prediction and report the comparison results with SOTA baseline models in Table 3. Our CenPre achieves overall better performance than the baseline models, reaching optimal performance in all datasets. This indicates that leveraging the node importance based on the notion of Centrality can improve the representations of nodes in a graph, thereby leading to better link prediction performance.

**Graph Classification** Table 4 shows the experimental results of graph classification. We can see that our CenPre consistently outperforms the baseline models on all datasets. This verifies the effectiveness of CenPre in graph classification, indicating that the proposed Centrality-guided method can improve the learning of the entire graph based on the improvement of node representations, therefore achieving optimal performance in graph classification. Further, our CenPre performs

Table 4: Experimental results of **graph classification**. Averaged accuracy±std. (%) over 10 runs are reported. The best and second-best results are highlighted in **red** and **blue**, respectively.

| | Methods | NCI1 | PROTEINS | DD | MUTAG | IMDB-B | RDT-B | A.R.↓ |
|---|---|---|---|---|---|---|---|---|
| Supervised | GCN | 76.29±1.79 | 75.17±3.63 | 73.26±4.46 | 79.81±1.58 | 57.35±4.04 | 81.30±6.93 | 12 |
| | GAT | 74.90±1.72 | 74.72±4.01 | 77.30±3.68 | 78.89±2.05 | 54.60±7.45 | 72.70±2.30 | 12.7 |
| | SAGE | 74.73±1.34 | 74.01±4.27 | 75.78±3.91 | 78.75±1.18 | 58.95±6.74 | 83.10±5.40 | 12.7 |
| Kernel | WL | 80.31±0.46 | 72.92±0.56 | 76.44±2.35 | 80.72±3.00 | 72.30±3.44 | 68.82±0.41 | 11.2 |
| | DGK | 81.01±1.06 | 73.30±0.82 | 74.85±0.74 | 87.44±2.72 | 66.96±0.56 | 78.04±0.39 | 10.3 |
| Self-supervised | node2vec | 54.89±1.61 | 57.49±3.57 | - | 72.63±10.20 | 56.40±2.80 | 69.70±4.10 | 16.2 |
| | Graph2Vec | 73.22±1.81 | 73.30±2.05 | 70.32±2.32 | 83.15±9.25 | 71.10±0.54 | 75.78±1.03 | 13.1 |
| | InfoGraph | 76.20±1.06 | 74.44±0.31 | 74.24±0.86 | 89.01±1.13 | 73.03±0.87 | 82.50±1.42 | 9.3 |
| | GraphCL | 77.87±0.41 | 74.39±0.45 | 78.62±0.40 | 86.80±1.34 | 71.14±0.44 | 89.53±0.84 | 8.2 |
| | JOAO | 78.07±0.47 | 74.55±0.41 | 77.32±0.54 | 87.35±1.02 | 70.21±3.08 | 85.29±1.35 | 9.2 |
| | InfoGCL | 80.20±0.60 | - | - | **91.20**±1.30 | 75.10±0.90 | - | 10.5 |
| | GraphMAE | 80.40±0.30 | 75.30±0.39 | - | 88.19±1.26 | **75.52**±0.66 | 88.01±0.19 | 7.5 |
| | SimGRACE | 79.12±0.44 | 75.35±0.09 | 77.44±1.11 | 89.01±1.31 | 71.30±0.77 | 89.51±0.89 | 6.3 |
| | AutoGCL | **82.00**±0.29 | 75.80±0.36 | 77.57±0.60 | 88.64±1.08 | 72.32±0.93 | 88.58±1.49 | 5 |
| | TopoGCL | 81.30±0.27 | **77.30**±0.89 | 79.15±0.35 | 90.09±0.93 | 74.67±0.32 | **90.40**±0.53 | **2.8** |
| | GPA | 80.42±0.41 | 75.94±0.25 | **79.90**±0.35 | 89.68±0.80 | 74.64±0.35 | 89.32±0.38 | 5.3 |
| | CenPre (ours) | **88.13**±0.91* | **80.25**±0.67* | **85.18**±1.37* | **94.74**±0.48* | **78.05**±1.21* | **91.20**±0.50 | **1.0** |

consistently better than TopoGCL (Chen et al., 2024) which considers the topological information of the graph. This denotes that leveraging node-level and graph-level importance can improve the learning of graph structure when modeling a graph, thus improving the performance of graph classification.

## 5.3 ANALYSIS OF OUR CENPRE

**Ablation Study**   We conduct an ablation study to analyze the impact of each module of the CenPre on performance. The results are reported in Table 5. We can see that whether using only node-level structure (w/ $\mathcal{L}_d$) or graph-level structure (w/ $\mathcal{L}_e$), the performance of the model has significantly decreased. This verifies that we need to simultaneously explore the different importance of nodes from both local and global perspectives in order to better learn the structural information of the graph. In ad-

Table 5: Experimental results of the ablation study. $\Delta$ denotes the performance drop relative to the full CenPre. "w/o SVD" means the adjacency matrix $\mathcal{A}$ is used directly as the structural representation, and AUC is used as the evaluation metric for Cora-Link.

| Methods | Cora-Node | $\Delta$ | Cora-Link | $\Delta$ | MUTAG-Graph | $\Delta$ |
|---|---|---|---|---|---|---|
| CenPre | 85.15±0.49 | 0.00 | 95.05±0.18 | 0.00 | 94.74±0.48 | 0.00 |
| w/ $\mathcal{L}_d$ | 81.43±0.30 | 3.72 | 92.80±1.05 | 2.25 | 91.62±0.05 | 3.12 |
| w/ $\mathcal{L}_e$ | 79.91±0.22 | 5.24 | 93.17±0.65 | 1.88 | 90.38±0.15 | 4.36 |
| w/o $\mathcal{L}_d$ | 80.40±0.10 | 4.75 | 94.08±0.29 | 0.97 | 86.84±0.13 | 7.9 |
| w/o $\mathcal{L}_e$ | 84.21±0.62 | 0.94 | 94.07±0.54 | 0.98 | 92.11±0.26 | 2.63 |
| w/o $\mathcal{L}_r$ | 78.63±0.50 | 7.12 | 91.29±1.57 | 3.76 | 89.47±2.55 | 5.27 |
| w/o SVD | 83.45±0.61 | 1.70 | 93.72±0.22 | 1.33 | 93.52±0.42 | 1.22 |

dition, the removal of graph representation alignment (w/o $\mathcal{L}_r$) seriously degrades the performance of our CenPre. This indicates that it is necessary to align the node representation when learning structural information, since it can prevent the loss of the graph semantic information, especially for node classification. Further, the removal of node-level importance learning (w/o $\mathcal{L}_d$) and graph-level importance learning (w/o $\mathcal{L}_e$) can lead to considerable performance degradation, which demonstrates that learning node importance from both node and graph levels can make full use of the important structural information of node in a graph, thus improving the model's performance. The performance of "w/o SVD" shows that the removal of SVD noticeably degrades the performance of our CenPre. This implies that exploiting SVD to produce the structure representation based on important singular values and eigenvectors can help the model better learn the graph's structural information.

**Analysis of Generalizability**   To analyze the generalizability of our CenPre to different graph autoencoders, we conduct comparative experiments based on different graph autoencoders and report the results in Table 6. We can see that the proposed CenPre can directly adapt to different graph autoencoders and achieve varying degrees of performance improvement, which validates the generalizability of our CenPre in

Table 6: Experimental results of using different graph autoencoders on the Cora dataset.

| Methods | Original | w/ CenPre | Improvement |
|---|---|---|---|
| GAE | 74.90±0.40 | 81.56±0.60 | 6.66 |
| VGAE | 76.30±0.20 | 78.30±1.15 | 2.00 |
| ARGA | 77.95±0.70 | 80.73±0.27 | 2.78 |
| DGI | 82.30±0.60 | 83.11±0.54 | 0.81 |
| CenPre (ours) | 85.15±0.49 | | 0.00 |

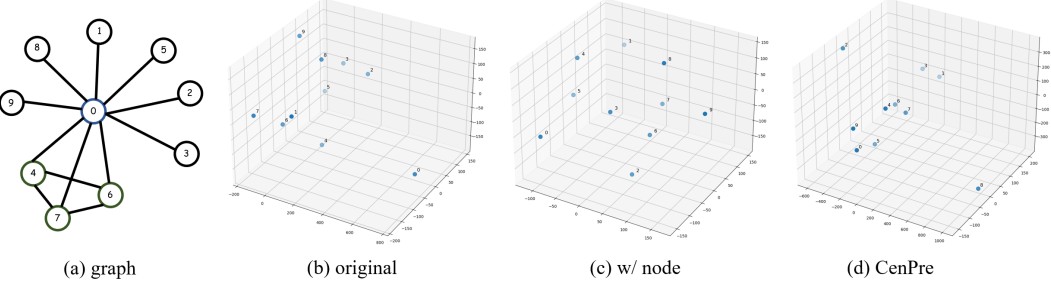

(a) graph    (b) original    (c) w/ node    (d) CenPre

Figure 2: The T-SNE 3D plots of the extracted nodes during different pre-training stages.

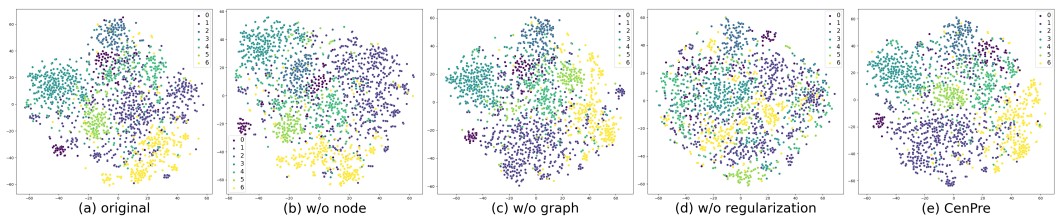

(a) original    (b) w/o node    (c) w/o graph    (d) w/o regularization    (e) CenPre

Figure 3: The T-SNE 2D plots of all the test samples from the CORA dataset.

graph pre-training. In addition, it can be noted that our CenPre, which uses the original node representation, achieves better performance than those that use graph autoencoders. This implies that using the original representation can better assist our CenPre in learning the structural information of the graph, thereby obtaining better node representations. Furthermore, this also indicates that exploring better methods for graph autoencoders may further enhance the learning of graphs.

**Analysis of Node Representation**    To analyze how our CenPre improves the representation of nodes, we select a node from Cora data and retain its first-order neighbors to analyze the changes in their representation at different learning stages. The results are shown in Figure 2. We can see that, the original representations of nodes are divergent and the correlation between them cannot be observed. After node-level importance learning, some important nodes are gathered together. Furthermore, through our complete CenPre, important nodes (4, 6, 7) are further aggregated (Figure 2 (d)). This indicates that our CenPre can learn the similarity importance between nodes based on Centrality, thereby improving the representation of nodes.

**Visualizations**    To qualitatively demonstrate how our CenPre improves the node representations, Figure 3 shows the t-SNE (van der Maaten & Hinton, 2008) visualization of node representations from original representations (a), the variants of our CenPre (b), (c) and (d), and our CenPre (e). We can observe that the original node representations largely diffuse and overlap between different labels. The variants of our CenPre can show differences between different labels, which denotes that exploring preferable methods to learn better node representations is key to improving node classification. Further, the node representations derived by our CenPre can be better separated from different labels. This indicates that our CenPre can make full use of the node importance based on Centrality, and further improve the representations of nodes based on the representation alignment, therefore leading to improved node representations.

## 6    CONCLUSION

In this paper, we propose **CenPre**, a **Cen**trality-guided **Pre**-training framework for node representation learning. CenPre integrates structural information into graph representation by using node importance based on Degree and Eigenvector Centrality while preserving semantic information through a graph alignment module. Experiments on real-world datasets demonstrate its superiority over state-of-the-art models in node classification, link prediction, and graph classification.

ACKNOWLEDGMENTS

We sincerely thank the anonymous reviewers for their insightful feedback and valuable suggestions, which have significantly enhanced the quality of this work. This work was partially supported by Hong Kong RGC GRF No. 14206324, CUHK direct grant No. 4055209, CUHK Knowledge Transfer Project Fund No. KPF23GWP20, National Natural Science Foundation of China 62176076, Natural Science Foundation of Guangdong 2023A1515012922, Shenzhen Foundational Research Funding JCYJ20220818102415032, and the Major Key Project of PCL2023A09, Guangdong Provincial Key Laboratory of Novel Security Intelligence Technologies 2022B1212010005, the UK Engineering and Physical Sciences Research Council through a New Horizons grant (EP/X019063/1) and by UKRI Impact Acceleration Accounts.

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

## A  APPENDIX

## A  CENTRALITY MEASURES

Centrality is a key concept in network analysis used to identify the most important or influential nodes within a graph. Different centrality measures capture various aspects of a node's importance, ranging from its immediate connections to its role in facilitating communication between other nodes. These measures help in understanding the structure of the graph, the flow of information, and the relative significance of individual nodes in maintaining the overall connectivity. Below, we introduce several commonly used centrality measures:

- **Degree Centrality** (Freeman, 1978) counts the number of direct connections a node has. Nodes with higher degree centrality are considered more locally important due to their numerous direct interactions within the network. For a node $v_i$, the degree centrality is given by:

$$C_D(v_i) = \sum_j \mathcal{A}_{ij} \tag{11}$$

where $\mathcal{A}_{ij}$ is the entry in the adjacency matrix $\mathcal{A}$, indicating the presence or absence of an edge between nodes $i$ and $j$.

- **Eigenvector Centrality** (Bonacich, 1972) assigns relative scores to nodes based on the principle that connections to high-scoring nodes contribute more to the score of the node itself. It captures both direct and indirect influences in the network. For a node $v_i$, the eigenvector centrality is given by:

$$C_E(v_i) = \frac{1}{\lambda} \sum_j \mathcal{A}_{ij} C_E(v_j) \tag{12}$$

where $\lambda$ is the largest eigenvalue of the adjacency matrix $\mathcal{A}$.

- **Betweenness Centrality** (Freeman, 1977) quantifies the number of times a node acts as a bridge along the shortest path between two other nodes. For a node $v_i$, betweenness centrality is given by:

$$C_B(v_i) = \sum_{s \neq v_i \neq t} \frac{\sigma_{st}(v_i)}{\sigma_{st}} \tag{13}$$

where $\sigma_{st}$ is the total number of shortest paths from node $s$ to node $t$, and $\sigma_{st}(v_i)$ is the number of those paths that pass through node $v_i$.

- **Closeness Centrality** (Bavelas, 1950; Sabidussi, 1966) measures how close a node is to all other nodes in the network. For a node $v_i$, closeness centrality is defined as:

$$C_C(v_i) = \frac{1}{\sum_j d(v_i, v_j)} \tag{14}$$

where $d(v_i, v_j)$ is the shortest path distance between nodes $v_i$ and $v_j$.

- **PageRank Centrality** is a variant of eigenvector centrality that evaluates the importance of a node based on the quality and quantity of incoming connections, where connections from more important nodes weigh more heavily. The PageRank score $PR(v_i)$ for a node $v_i$ is given by:

$$PR(v_i) = \frac{1-d}{N} + d \sum_{j \in \mathcal{N}(v_i)} \frac{PR(v_j)}{|\mathcal{N}(v_j)|} \tag{15}$$

where $d$ is the damping factor (typically set to 0.85), $N$ is the total number of nodes, and $\mathcal{N}(v_j)$ is the set of neighbors of node $v_j$.

- **Katz Centrality** (Katz, 1953) extends degree centrality by considering the total number of walks between nodes. It is defined as:

$$C_K(v_i) = \alpha \sum_j \mathcal{A}_{ij} C_K(v_j) + \beta \tag{16}$$

where $\alpha$ is a constant (decay factor) and $\beta$ is an additional weight applied to each node, allowing for paths of different lengths to contribute to the centrality.

- **Harmonic Centrality** (Marchiori & Latora, 2000) is a variation of closeness centrality that computes the sum of the reciprocals of the shortest path distances from one node to all other nodes. It is defined as:

$$C_H(v_i) = \sum_{j \neq i} \frac{1}{d(v_i, v_j)} \tag{17}$$

where $d(v_i, v_j)$ is the shortest path distance between nodes $v_i$ and $v_j$.

Table 7 provides a detailed comparison of node types based on Degree Centrality and Eigenvector Centrality, highlighting their roles and characteristics in network structures. The table categorizes nodes into four distinct types depending on their local and global importance, as measured by high or low values of these centrality metrics. Each category is accompanied by a description of the node's structural characteristics within the network and an illustrative example from social networks, offering a practical understanding of how centrality measures reflect different types of influence and connectivity in real-world scenarios.

## B EIGEN-DECOMPOSITION (ED) AND SINGULAR VECTOR DECOMPOSITION (SVD)

**Theorem B.1.** *ED and SVD of a matrix $\mathcal{A}$ are equivalent if and only if $\mathcal{A}$ is symmetric positive semi-definite, i.e., $\mathcal{A}^{\mathsf{T}} = \mathcal{A} \wedge \mathcal{A} \succeq 0$.*

*Proof.* Given a matrix $\mathcal{A} \in \mathbb{R}^{n \times n}$, the SVD of $\mathcal{A}$ is written as:

$$\mathcal{A} = U\Sigma V^{\mathsf{T}} \tag{18}$$

where:

| Degree | Eigen | Node Characteristics | Examples in Social Network |
|--------|-------|----------------------|----------------------------|
| High | Low | A node connected to many other nodes that are not highly central. Important locally but not globally. | A celebrity with many followers but most of those followers are inactive or non-influential accounts. |
| Low | High | A node with few direct connections but connected to highly central nodes. Important globally but not locally. | A journalist or analyst who is followed by a few highly influential public figures or news organizations. |
| High | High | A node that is both locally and globally important, with many connections and highly central neighbors. | A popular politician who has many followers who are themselves influential, such as other public figures. |
| Low | Low | A node with few connections and low global importance. Neither locally nor globally important. | A regular person with few followers, none of whom are influential or central. |

Table 7: Comparison of Degree Centrality and Eigenvector Centrality for Different Node Types and their examples in social networks.

- $U \in \mathbb{R}^{n \times n}$ is an orthogonal matrix (i.e., $U^\mathsf{T} U = I$),

- $V \in \mathbb{R}^{n \times n}$ is an orthogonal matrix (i.e., $V^\mathsf{T} V = I$),

- $\Sigma \in \mathbb{R}^{n \times n}$ is a diagonal matrix containing the singular values $\sigma_1, \sigma_2, \ldots, \sigma_n$, where $\sigma_i \geq 0$.

The ED of a matrix $\mathcal{A} \in \mathbb{R}^{n \times n}$ is given by:

$$\mathcal{A} = Q \Lambda Q^{-1} \tag{19}$$

where:

- $Q \in \mathbb{R}^{n \times n}$ is an invertible matrix whose columns are the eigenvectors of $\mathcal{A}$,

- $\Lambda \in \mathbb{R}^{n \times n}$ is a diagonal matrix whose entries are the eigenvalues $\lambda_1, \lambda_2, \ldots, \lambda_n$.

We want to establish the conditions under which the SVD and ED of a matrix $\mathcal{A}$ are equivalent. Specifically, we need to check under what conditions:

$$U \Sigma V^\mathsf{T} \equiv Q \Lambda Q^{-1} \tag{20}$$

For the two decompositions to be equivalent, the matrix $\mathcal{A}$ must be symmetric, which implies that the left and right singular vectors are the same:

$$\mathcal{A} = \mathcal{A}^\mathsf{T} \to \mathcal{A} = U \Sigma U^\mathsf{T} \tag{21}$$

In SVD, the singular values $\sigma_i$ in $\Sigma$ are always non-negative. For the eigenvalues $\lambda_i$ in the ED to match the singular values, all eigenvalues must be non-negative as well. This requires that $\mathcal{A}$ be **positive semi-definite**. For SVD and ED to coincide, the matrix $Q$ in the ED must be orthogonal, i.e., $Q^\mathsf{T} Q = I$. This occurs when the eigenvectors of $A$ are orthonormal. In this case, $Q^{-1} = Q^\mathsf{T}$, and the ED becomes:

$$\mathcal{A} = Q \Lambda Q^\mathsf{T} \tag{22}$$

Thus, for $\mathcal{A}$ to have orthonormal eigenvectors, it must be symmetric.
In conclusion, the SVD and ED of a matrix $\mathcal{A}$ are equivalent if and only if:

- $\mathcal{A}$ is symmetric, i.e., $\mathcal{A} = \mathcal{A}^\mathsf{T}$,

- $\mathcal{A}$ is positive semi-definite,

- The eigenvectors of $\mathcal{A}$ are orthonormal.

In this case, the SVD and ED both yield the same decomposition:

$$U\Sigma U^{\mathsf{T}} \equiv Q\Lambda Q^{\mathsf{T}} \tag{23}$$

where $\Sigma = \Lambda$, and the columns of $U$ (in SVD) and $Q$ (in ED) are the same orthonormal eigenvectors of $\mathcal{A}$. Thus, **SVD and ED are equivalent when $\mathcal{A}$ is symmetric and positive semi-definite**. $\qquad\square$

## C BASELINES

We compare and evaluate our CenPre framework with a series of baseline models, which are grouped by graph kernel methods, graph supervised learning methods, and graph self-supervised learning methods. The more detailed introduction of the baseline models is as follows:

**Graph Kernels Methods:**

- **WL** (Shervashidze et al., 2011) measures graph similarity by iteratively refining node labels through aggregation of neighboring node labels, capturing graph structure and preserving node attribute information.
- **DGK** (Yanardag & Vishwanathan, 2015) uses neural networks to learn representations of subgraphs and combine them to compute a similarity score, enabling more expressive and flexible graph comparisons for various graph analysis tasks.

**Graph Supervised Learning Methods:**

- **GCN** (Kipf & Welling, 2017) uses spectral graph theory to aggregate neighboring node features, learning new node representations for tasks like classification and link prediction.
- **GAT** (Velickovic et al., 2018) uses an attention mechanism to assign weights to neighboring nodes, computing weighted sums of their features to capture complex relationships for tasks where neighbor importance varies.
- **GIN** (Xu et al., 2019) aggregates node features by summing a node's features with its neighbors' and applying an MLP, capturing local structure and complex feature interactions for effective node and graph classification.
- **SAGE** (Hamilton et al., 2017) generates node embeddings by sampling and aggregating neighborhood features, enabling it to generalize to unseen nodes and graphs for scalable node classification and link prediction.

**Graph Self-supervised Learning Methods**

- **GAE** (Kipf & Welling, 2016) encodes nodes into latent embeddings and reconstructs the graph structure to learn meaningful representations for tasks such as node classification and link prediction.
- **VGAE** (Kipf & Welling, 2016) extends the GAEs with variational inference to learn probabilistic latent variable models, providing robust and expressive node embeddings for graph-based tasks such as link prediction and anomaly detection.
- **ARGA** (Pan et al., 2018) enhances GAEs by incorporating adversarial training, ensuring more robust and discriminative node embeddings for graph tasks like node classification and link prediction through adversarial regularization.
- **GraphMAE** (Hou et al., 2022) leverages masked node feature reconstruction to learn rich and informative node embeddings, enhancing performance on downstream graph tasks such as node classification and graph classification.
- **MaskGAE** (Li et al., 2023) applies masking and reconstruction strategies to learn meaningful graph representations, enhancing performance on tasks such as node classification and link prediction.
- **DGI** (Veličković et al., 2019) maximizes mutual information between local and global graph representations, producing highly expressive node embeddings that excel in tasks such as node classification and graph classification.

- **GRACE** (Zhu et al., 2020) leverages contrastive learning to maximize agreement between different views of the same graph, resulting in robust and informative node embeddings for various graph-based tasks.

- **GCA** (Zhu et al., 2021b) enhances contrastive learning by incorporating adaptive augmentation strategies, yielding discriminative and robust node embeddings for diverse graph-related tasks.

- **BGRL** (Thakoor et al., 2021) learns graph representations by leveraging a bootstrapping mechanism to predict target network outputs, producing effective node embeddings for tasks such as node classification and link prediction without the need for negative sampling.

- **CCA-SSG** (Zhang et al., 2021) utilizes canonical correlation analysis to maximize agreement between different graph views, generating high-quality node embeddings for downstream tasks such as node classification and clustering.

- **node2vec** (Grover & Leskovec, 2016) uses biased random walks to generate node sequences, which are then used with the Skip-gram model to produce continuous node embeddings for downstream tasks like node classification and link prediction.

- **graph2vec** (Narayanan et al., 2017) uses the Skip-gram model to generate continuous embeddings, capturing structural and global graph characteristics for downstream tasks.

- **InfoGraph** (Sun et al., 2020) maximizes mutual information between graph-level and substructure-level representations, resulting in informative embeddings for downstream graph classification and clustering tasks.

- **GraphCL** (You et al., 2020) designs four contrastive learning with graph augmentations to capture rich and robust graph representations, improving performance on tasks such as graph classification and clustering.

- **JOAO** (You et al., 2021) dynamically optimizes data augmentations to enhance contrastive learning, producing robust and generalizable node embeddings for various graph tasks.

- **InfoGCL** (Xu et al., 2021) combines contrastive learning with mutual information maximization to enhance node and graph-level embeddings, yielding improved performance on various graph-based tasks.

- **Patcher** (Ju et al., 2023) mitigates degree bias in Graph Neural Networks through test-time augmentation, enhancing the robustness and generalization of graph representations for various graph-based tasks.

- **SimGRACE** (Xia et al., 2022a) enhances graph contrastive learning by leveraging similarity-based augmentations, producing robust node and graph embeddings for improved performance on downstream tasks.

- **AutoGCL** (Yin et al., 2022) employs learnable view generators to create optimized augmentations, leading to robust and effective graph embeddings for various downstream tasks.

- **TopoGCL** (Chen et al., 2024) leverages topological invariance and extended persistence to capture higher-order substructures, enhancing graph representations and delivering significant performance gains in unsupervised graph classification, particularly in biological, chemical, and social interaction graphs.

- **GPA** (Zhang et al., 2024) customizes augmentation strategies for each graph based on its topology and node attributes, enhancing representation learning and outperforming state-of-the-art models across diverse benchmark datasets.

- **NodeProperty** (Jin et al., 2020) combines tasks such as clustering and node distance prediction to learn robust structural embeddings.

## D   MORE DETAILS OF EXPERIMENTAL IMPLEMENTATION

For node classification and link prediction tasks, we use a 2-layer GCN (Kipf & Welling, 2017) as the encoder for self-supervised pretraining under the CenPre framework, increasing to 4 layers for the arXiv (Hu et al., 2020) dataset. After pretraining, we freeze the encoder to extract node embeddings, which are then input into a 2-layer MLP for classification and prediction. Results are

Table 8: Experimental implementation for node classification, link prediction, and graph classification. In node classification tasks, we set Hidden Size to 64 for the Citation Networks, 128 for the Amazon Co-Buy, and 512 for the arXiv dataset, which has a 4-layer MLP as the downstream classifier.

| Parameter | Description | Node Class. | Link Pred. | Graph Class. |
|---|---|---|---|---|
| $LR$ | Learning Rate | 0.001 | 0.001 | 0.001 |
| $L_2$ | Weight Decay | 5e-4 | 5e-4 | 5e-4 |
| $p_e$ | Early Stopping Patience | 15 | 15 | 15 |
| $e_\Delta$ | Early Stopping Min-Delta | 1e-5 | 1e-5 | 1e-5 |
| $p_s$ | Learning Rate Scheduler Patience | 8 | 8 | 8 |
| $\mathcal{B}$ | Batch size | 128 | 128 | 1 |
| $K$ | Number of GNN Layers | 2 | 2 | 2 |
| $M$ | Number of MLP Layers | 2/4 | 2 | 2 |
| $d_h$ | Hidden Layer Size | 64/128/512 | 64 | 128 |
| $\lambda_1$ | Scaling Weight of Node-Level Loss $\mathcal{L}_{node}$ | 1 | 1 | 1 |
| $\lambda_2$ | Scaling Weight of Graph-Level Loss $\mathcal{L}_{graph}$ | 1 | 1 | 1 |
| $\lambda_3$ | Scaling Weight of Regularization Loss $\mathcal{L}_{reg}$ | 5 | 5 | 5 |
| $E$ | Training epochs | 100 | 100 | 300 |
| $\eta_d$ | Dropout Ratio | 0.2 | 0.2 | 0.2 |
| $\text{POOL}(\cdot)$ | Pooling Function | - | - | mean |

reported as mean and standard deviation over 5 runs. For graph classification, we use a GIN (Xu et al., 2019) encoder instead, which is commonly used in previous graph classification works. we use the Adam (Kingma & Ba, 2014) optimizer with an initial learning rate of $lr = 0.01$. The balancing hyperparameters for loss components are set to $\lambda_1 = 1$, $\lambda_2 = 1$, and $\lambda_3 = 5$, determined through pilot studies. For model parameters, we use grid search to find the optimal parameter combination. For the three hyperparameters $\lambda_1$, $\lambda_2$, and $\lambda_3$, in preliminary experiments, we found that when these three hyperparameters are within a reasonable range of values, the performance of the model will only fluctuate within a certain range. When hyperparameters are set to $\lambda_1 = 1$, $\lambda_2 = 1$, and $\lambda_3 = 5$, the model's performance is optimal. Therefore, we set $\lambda_1 = 1$, $\lambda_2 = 1$, and $\lambda_3 = 5$, also making them to the same scale.

Our framework is built on PyTorch (Paszke et al., 2019) and PyTorch Geometric (Fey & Lenssen, 2019), leveraging their datasets and functionalities. Experiments are conducted on an Intel(R) Xeon(R) Gold 6248R CPU at 3.00GHz and an Nvidia Tesla V100 GPU with 32GB VRAM. Table 8 shows all relevant parameter settings for our experiments.

# E COMPLEXITY ANALYSIS

To analyze the computational requirements of CenPre, we consider the complexity of its iterative operations during training while treating the truncated-SVD as a pre-computation step. The precomputed structural representations are reused throughout the training process, and their computation does not contribute to the iterative complexity of the framework.

## E.1 SPACETIME COMPLEXITY OF CENPRE

The space complexity of CenPre during training is determined by:

- The node embeddings, are stored as a matrix of size $O(nd)$, where $n$ is the number of nodes and $d$ is the dimensionality of the embeddings.

- The adjacency matrix $\mathcal{A}$, which has $O(|E|)$ space complexity for sparse graphs, where $|E|$ is the number of edges.

- The cross-attention mechanism, which requires storing $O(n^2)$ attention weights during alignment.

Combining these, the overall space complexity of CenPre is:

$$O(nd + |E| + n^2) \tag{24}$$

The time complexity of CenPre's iterative operations during training is as follows:

- Node-level importance learning: Computing node degrees from the adjacency matrix $\mathcal{A}$ has $O(|E|)$ complexity. Training the degree predictor $\mathcal{P}_d$ for $n$ nodes requires $O(nd)$ operations per iteration.

- Graph-level importance learning: The cross-attention mechanism aligns graph and structural representations, incurring $O(n^2d)$ complexity due to pairwise attention calculations.

- Graph representation alignment: Computing the $L_2$-norm alignment between structure-fused and original embeddings requires $O(nd)$ operations.

The total time complexity per training iteration is:

$$O(|E| + nd + n^2d) \tag{25}$$

Note that the truncated-SVD operation is considered a preprocessing step and involves decomposing the adjacency matrix $\mathcal{A}$ to obtain the top $k$ singular vectors. Its time complexity is $O(kn^2)$, and space complexity is $O(kn)$, where $k$ is the number of retained singular values. This cost does not contribute to the complexity of iterative training and is incurred only once.

### E.2 TIME COMPLEXITY COMPARISON WITH OTHER METHODS

In this subsection, we provide a concise summary of the theoretical time complexity of other methods.

- **GraphCL** costs $O(L|E|d^2 + nd^2 + N^2d)$ per iteration. This includes $O(L|E|d^2)$ for the GNN encoder over $L$ layers, $O(nd^2)$ for the projection head, and $O(N^2d)$ for contrastive loss computation with minibatch size $N$. Graph augmentation adds minor costs ($O(n)$ to $O(|E_s|)$).

- **GraphMAE** consists of three main components. Masked feature reconstruction, including masking and re-masking, incurs $O(n)$ complexity for $n$ nodes. The GNN encoder and decoder operate over $L$ layers with a complexity of $O(L|E|d^2)$, where $|E|$ is the number of edges and $d$ is the embedding dimension. The scaled cosine error computation for feature reconstruction adds $O(nd)$. Thus, the overall time complexity per iteration is $O(L|E|d^2 + nd)$.

- **MaskGAE** per iteration includes $O(L|E_{\text{vis}}|d^2)$ for the GNN encoder processing the unmasked graph over $L$ layers, $O(|E_{\text{mask}}|d)$ for the structure decoder reconstructing masked edges, and $O(nd)$ for the degree decoder approximating node degrees, where $|E_{\text{vis}}|$ and $|E_{\text{mask}}|$ are the numbers of unmasked and masked edges, respectively. Overall, the complexity is $O(L|E_{\text{vis}}|d^2 + |E_{\text{mask}}|d + nd)$.

- **TopoGCL** includes three main components. Extended persistence computation, involving simplicial complex operations, has a complexity of $O(N^3)$ for $N$ nodes. The GNN encoder, applied over $L$ layers, incurs $O(L|E|d^2)$ for $|E|$ edges and embedding dimension $d$. Contrastive loss computation for a minibatch of size $M$ adds $O(M^2d)$. Thus, the overall complexity is $O(N^3 + L|E|d^2 + M^2d)$, with extended persistence dominating for dynamic graphs.

- AutoGCL incurs $O(L|E|d^2)$ for the GNN-based view generator and encoder operating over $L$ layers with $|E|$ edges and embedding dimension $d$, and $O(N^2d)$ for contrastive loss computation with minibatch size $N$. Thus, the overall complexity is $O(L|E|d^2 + N^2d)$.

The time complexity comparison highlights the computational efficiency and scalability of CenPre compared to other graph learning methods. CenPre achieves a lower overall complexity of $O(|E| + nd + n^2d)$ by leveraging precomputation for structural representations, making it well-suited for large sparse graphs. In contrast, methods like GraphCL and AutoGCL incur higher costs due to the quadratic dependence on minibatch size ($O(N^2d)$) in contrastive loss computation, which can become a bottleneck for large-scale data. GraphMAE and MaskGAE share similar complexities, but their dependency on the number of masked and unmasked edges ($|E_{\text{vis}}|$ and $|E_{\text{mask}}|$) introduces sensitivity to the masking ratio. TopoGCL stands out with a cubic dependence on the number of nodes ($O(n^3)$) for extended persistence computation, making it less efficient for large graphs, though it provides rich topological insights. Overall, CenPre balances computational demands with performance by focusing on centrality measures and pretraining strategies, offering a practical advantage in terms of scalability and efficiency.

# F    SCALING SENSITIVITY V.S. RANKING SENSITIVITY

Another significant challenge in the centrality-guided pre-training method is its sensitivity to scaling. As noted by (Hu et al., 2019a), centrality scores are not directly comparable across graphs of varying scales. Consequently, focusing solely on the centrality-based loss may fail to effectively capture meaningful features.

To tackle the issue of scaling sensitivity, some existing studies have proposed learning the centrality-based ranking instead of directly optimizing an explicit centrality loss (Hu et al., 2019a). However, a slight modification to a graph, such as the removal of a few edges, may significantly alter centrality values without necessarily impacting the ranking.

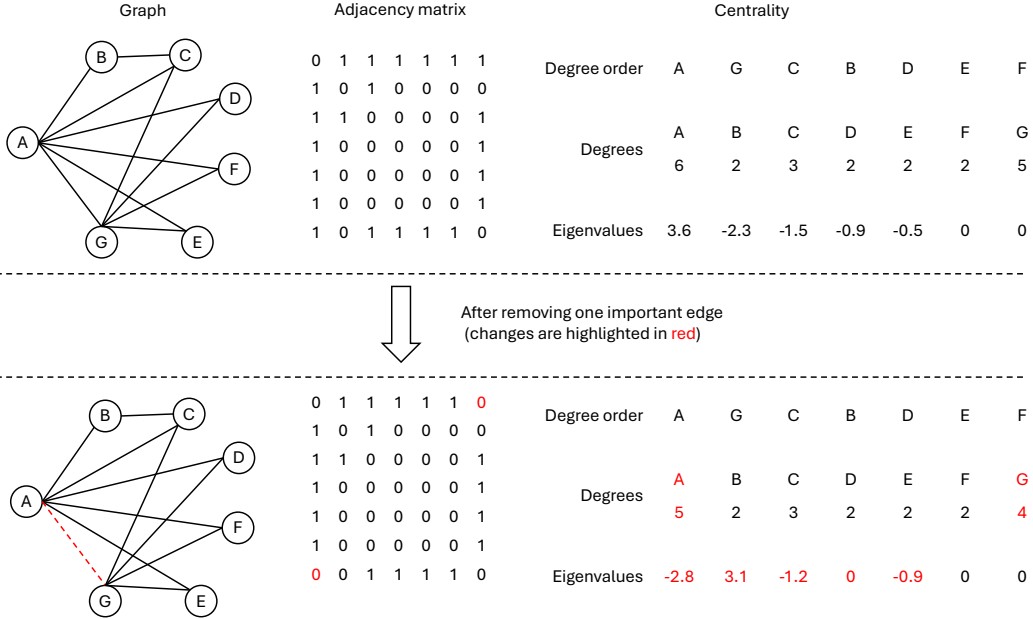

Figure 4: The changes of different centrality after removal of one edge $\overline{AG}$. We highlight the changes in red color, where you could find a minor removal might cause significant changes in many different centralities but cannot be captured by the order-based feature, which is proposed in (Hu et al., 2019a) to reduce the scaling sensitivity but might introduce a new problem.

Here, we demonstrate this property using a simple example graph with 7 nodes (as shown in Figure 4). When the edge between node A and node G is removed, the degree of both nodes A and G changes, but their rankings based on degree centrality remain unchanged. This illustrates that relying solely on degree order-based centrality fails to capture structural changes in the graph, as evidenced by the significant updates in eigenvalues following the edge removal.

For a similar reason, in a graph containing several clusters of nodes with only a single edge connecting different clusters, removing the connecting edges between clusters may leave the ranking-based loss unchanged but fundamentally alters the semantic structure of the nodes (as shown in Figure 5). Consequently, training strategies designed to address scaling sensitivity, such as ranking-based loss, may inadvertently introduce sensitivity to ranking changes.

Therefore, in our work, we address this issue by employing a representation alignment-based loss, as defined in Equation 9, to ensure that the learned representations align with an existing reference scale. This approach can be intuitively understood as aligning the learned representations with a prior distribution to maintain an appropriate scale. By introducing the parameter $\lambda_3$, which controls the contribution of $\mathcal{L}_r$ in the final loss, we strike a balance between scaling sensitivity and ranking sensitivity.

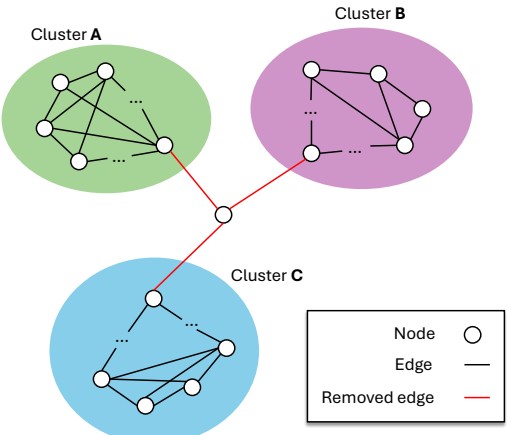

Figure 5: A graph which contains several clusters of nodes, where the nodes are almost fully connected within clusters but only one edge connected between different clusters. Removing the connecting edges (red edges) between clusters might not change the ranking-based loss but completely change the semantic structure of nodes.

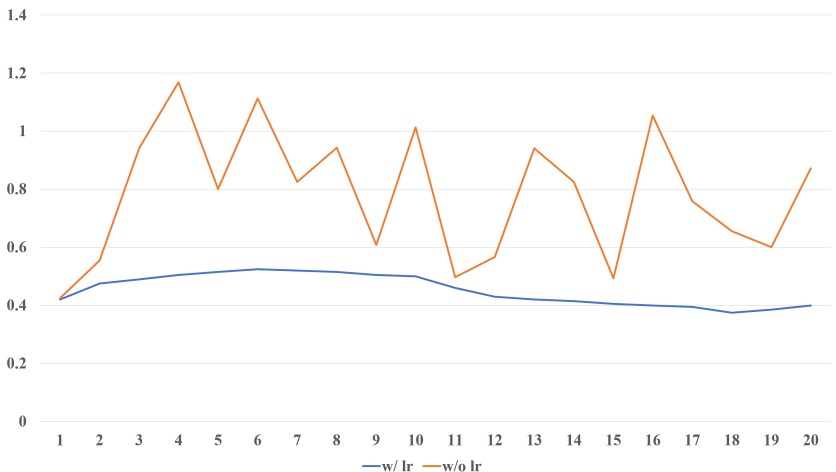

Figure 6: Comparison results of using and not using graph representation alignment on the Cora dataset. "lr" represents graph representation alignment.

To investigate the efficiency of the proposed method, we consider three dimensions to evaluate the proposed module:

- Does the proposed alignment strategy lead to a better performance?
- Does the proposed alignment strategy wildly be used for many different node representations prior to distribution?
- Does the proposed alignment strategy lead to a more stable training process under the perspective of scaling?

For the first question, compared to (Hu et al., 2019a), which employs a ranking-based loss to address scaling sensitivity, our proposed method demonstrates the ability to effectively capture centrality-guided information using an alignment-based loss, as shown in Table 5. Notably, the removal of the alignment-based loss results in a significant performance drop compared to other components, highlighting its importance.

For the second question, as shown in Table 6, the node encoding results can be utilized as prior knowledge within the alignment-based loss to consistently enhance performance.

For the third question, we conduct an additional experiment to track the $L_2$-norm variations during the training process. If the proposed method fails to address the scaling sensitivity issue fully, the learned norm would exhibit rapid fluctuations during optimization. In this experiment, we calculate and compare the average $L_2$-norm of all nodes with and without graph representation alignment, as shown in Figure 6. In the figure, without graph representation alignment, the average $L_2$-norm fluctuates significantly within the range $[0.4, 1.2]$ over 20 training epochs. This behavior suggests that the model struggles to appropriately scale and balance the learning objectives within a unified space when directly targeting scaling-sensitive objectives. Conversely, with our proposed alignment strategy, the learned representations stabilize around a consistent norm of $0.4$. It indicates that our alignment strategy effectively addresses the scaling sensitivity issue without requiring ranking-based learning objectives, which introduce the ranking sensitivity issue and potentially harm the performance.

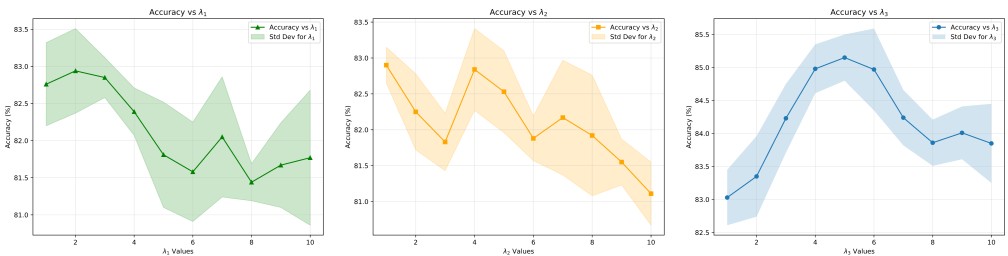

Figure 7: Performance trends of the CenPre on the Cora dataset as a function of hyperparameters $\lambda_1$, $\lambda_2$, and $\lambda_3$. Each graph shows the model's accuracy (%) as the corresponding hyperparameter is varied while keeping the other two fixed at a value of 1. The shaded bands around each curve represent the standard deviation, highlighting the variability in performance.

## G  ANALYSIS OF THE HYPERPARAMETERS

Figure 7 demonstrates the sensitivity of the CenPre's performance to the variation of hyperparameters $\lambda_1$, $\lambda_2$, and $\lambda_3$, which control the contributions of the alignment, eigenvalue, and degree losses, respectively. The results show that the performance of the model will only fluctuate within a certain range when the values of hyperparameters are set within a reasonable range. For example, when the value range of $\lambda_1$ is 1-4, the fluctuation amplitude of model performance is within 1%. Further, we can see that both excessively large and excessively small values can lead to a clear decrease in performance. In addition, analysis reveals that the model is most sensitive to $\lambda_3$, showcasing the critical role of the alignment loss in optimizing performance. Based on the results of the three hyperparameters, we can conclude that the model performs best when $\lambda_1 = 1$, $\lambda_2 = 1$, and $\lambda_3 = 5$. One possible reason is that this setting places the three hyperparameters on the same scale, which can lead to better learning of losses. Therefore, we set $\lambda_1 = 1$, $\lambda_2 = 1$, and $\lambda_3 = 5$ in our experiments.

## H  DISCUSSION REGARDING THE TRAINING STABILITY

In the proposed framework, there are three types of operations involved:

**Definition 3.** *Operation-1 Interaction between two trainable vectors, such as the inner product between a weight matrix/vector and node representations.*

**Definition 4.** *Operation-2 A feed-forward layer with vectorized inputs and outputs.*

**Definition 5.** *Operation-3 Interaction between predictive vectors and optimization objectives.*

Accordingly, the discussion will contain three parts:

**Proposition H.1.** *If the two vectors in **Operation-1** are independently drawn from Gaussian distributions, then the output of **Operation-1** also follows a Gaussian distribution.*

**Proposition H.2.** *If the input of **Operation-2** is Gaussian, then with high probability, the output distribution of **Operation-2** remains bounded.*

**Proposition H.3.** *Based on the assumptions of* ***Proposition*** *H.1 and* ***Proposition*** *H.2, the difference in gradients between any two training samples defined in* ***Operation-3*** *is approximately bounded.*

**Proposition** H.1 and **Proposition** H.2 are straightforward and intuitive.

## H.1 PROOF OF PROPOSITION H.1

*Proof.* Let the representation vector of a node be $r = \{r_1, r_2, ..., r_n\}$ and the weight matrix be $W = w_{ij}$, where $i = 1, 2, ..., n$ and $j = 1, 2, ..., k$. Assume they are randomly initialized as:

$$r_i \sim \mathcal{N}(0, \sigma_r^2), w_i \sim \mathcal{N}(0, \sigma_w^2). \tag{26}$$

By the normal product distribution, each term $w_{ij}r_i$ has a mean of zero and a variance proportional to the product of the variances:

$$\mathbb{E}(w_{ij}r_i) = 0, \ Var(w_{ij}r_i) = \sigma_r^2 \sigma_w^2. \tag{27}$$

For the output of **Operation-1**, defined as $z = Wr$, where $z = z_1, z_2, ..., z_k$ and $z_j = \Sigma_{i=1}^n w_{ij}r_i$, we obtain:

$$z_j \sim \mathcal{N}(0, n\sigma_r^2\sigma_w^2). \tag{28}$$

Thus, for any $\epsilon \geq 0$, there exists $\delta \geq 0$ such that

$$P(||z|| > \delta) \leq \epsilon. \tag{29}$$

In addition, applying a layer normalization operator after **Operation-1** can further constrain the variance.

$\square$

## H.2 PROOF OF PROPOSITION H.2

*Proof.* By the Heine-Cantor theorem, all mapping functions in **Operation-2** are continuous on $\mathcal{R}^n$. Since the input is Gaussian and bounded with high probability, the output is also bounded. This property can be described using Lipschitz continuity: for any pair of inputs $z_a$ and $z_b$ in **Operation-2**, there is a constant $K$ that bounds the corresponding output $O_a$ and $O_b$ by:

$$||O_a - O_b|| \leq K||z_a - z_b||. \tag{30}$$

Thus, we have:

$$\frac{\partial O_a}{\partial z_a} \leq K, \tag{31}$$

if we force $z_b$ approximate to $z_a$.

$\square$

## H.3 PROOF OF PROPOSITION H.3

*Proof.* By probabilistic convergence, for any $\epsilon \geq 0$, there exist $\delta \geq 0$, and a content $K \geq 0$ such that for any pair of inputs $r_a$ and $r_b$, after applying **Operation-1** and **Operation-2**, the output satisfied:

$$||O_a - O_b|| < K\delta, \text{ with probability } 1 - \epsilon. \tag{32}$$

In our training step, each node will be updated $n$ times, once as a positive sample and $n-1$ times as a negative sample. The objective label is based on the index, ensuring $||\mathcal{Y}_i^e|| = 1$.

Denoting $\mathcal{P}_e(r_i; \theta_e)$ as the function formed by **Operation-1** and **Operation-2**, and using the notation $O_a$ and $z_a$ for any input representation $r_a$, we derive:

$$\mathcal{Y}_a^e \frac{1}{O_a} \frac{\partial O_a}{\partial z_a} \frac{\partial z_a}{\partial r_a} \leq \frac{WK}{O_a}. \tag{33}$$

For a specific dimension $r_{ai}$, the gradient update is:

$$\mathcal{Y}_a^e \frac{1}{O_a} \frac{\partial O_a}{\partial z_a} \frac{\partial z_a}{\partial r_{ai}} \leq \mathcal{Y}_a^e \frac{K}{O_a} \sum_{j=1}^{k} w_{ij}. \tag{34}$$

Since $\sum_{j=1}^{k} w_{ij}$ follows a normal distribution due to independent initialization, we obtain a similar gradient for another sample $r_b$:

$$\mathcal{Y}_b^e \frac{1}{O_b} \frac{\partial O_b}{\partial z_b} \frac{\partial z_b}{\partial r_{bi}} \leq \mathcal{Y}_b^e \frac{K}{O_b} \sum_{j=1}^{k} w_{ij}. \tag{35}$$

Considering $||\mathcal{Y}_a^e||$ and $||\mathcal{Y}_b^e||$ are all identifiable representations of labels (only one dimension is 1 and others are $o$), we then have $||\mathcal{Y}_a^e|| = ||\mathcal{Y}_b^e|| = 1$. In addition, $O_a$ and $O_b$ are predictive outputs, which are usually normalized by activation such as softmax.

For a specific dimension, we conclude that the gradients for updating should remain on the same scale. Moreover, each training sample is treated as a positive sample once and as a negative sample $n-1$ times. This ensures that the updates to the representations are uniformly scaled across training samples and identifiable labels. In contrast, the learned representations may become biased when predicting non-identifiable labels or dealing with imbalanced category types. As highlighted in (Hu et al., 2019b), directly optimizing the eigenvalues can yield negative outcomes, as it may be initially caused by the imbalanced distribution in $\mathcal{Y}_b^e$ and subsequently in $O_i$.

$\square$

## I    LIMITATIONS

Despite its advantages, the CenPre framework has certain limitations. Firstly, its reliance on centrality measures may not fully capture complex, multi-faceted structural properties of graphs, potentially leading to sub-optimal representations in dense graphs and heterogeneous networks. Additionally, the computational complexity of calculating centrality measures, especially for large-scale graphs, can be high, impacting scalability and efficiency. Furthermore, the framework's performance may be sensitive to the choice of centrality measures, requiring careful tuning and selection based on specific graph characteristics. Future work should address these limitations by exploring more comprehensive structural descriptors, optimizing computational efficiency for broader applicability, and trying to explore an effective method to exploit the node importance in a more differentiated way.

