# OpenReview forum: "Centrality-guided Pre-training for Graph"
_ICLR.cc/2025/Conference — ICLR 2025 Poster_

### Official Review · Reviewer_eSwp · 2024-10-30

**Soundness:** 2
**Presentation:** 2
**Contribution:** 2
**Rating:** 6
**Confidence:** 3

**Summary:**

To address the problem that existing graph SSL methods focus on augmentation but overlook the integration of structural information, this paper proposes a Centrality-guided Graph Pre-training (CenPre) framework to learn better graph representations that incorporate structural information. CenPre consists of three modules that use degree, eigenvector centrality, and original graph representations to conduct aligning during pre-training. Experimental results demonstrate the effectiveness of CenPre on various tasks and datasets.

**Strengths:**

1. The experiments are comprehensive. A large number of datasets from various tasks are included, validating the effectiveness of the proposed method.
2. This paper is easy to read.

**Weaknesses:**

1. **The motivation is not well explained.**

    As the foundation of this paper, the authors claim that existing graph SSL methods only focus on augmentation and neglect the integration of structural information. This is somewhat not convincing. a) It is well known that GNNs can effectively capture both semantic and structural information simultaneously, especially considering some GNN variants with stronger expressive power. It is arbitrary to say that existing SSL methods fail to incorporate structural information because they can implicitly achieve this. b) At the same time, it is also common practice to explicitly consider structural information in existing SSL works. More theoretical or experimental evidence should be provided regarding the limitations of existing methods.

2. **The method lacks novelty.**

    The proposed CenPre basically uses centrality to guide the representation learning during pre-training. However, the concept of centrality has already been widely introduced in GNNs [1,2,3,4,5]. Among them, [4,5] exactly use centrality as a pretext task for pre-training. A detailed discussion on the differences between CenPre and previous works is necessary, which is missing in this paper.


Minor Point:

The paper uses a considerable amount of space (Sec 4.1 and Sec 4.2) to introduce the well-known concept of centrality, which is unnecessary.

[1] Maurya, Sunil Kumar, Xin Liu, and Tsuyoshi Murata. "Fast approximations of betweenness centrality with graph neural networks." *Proceedings of the 28th ACM international conference on information and knowledge management*. 2019.

[2] Avelar, Pedro, et al. "Multitask learning on graph neural networks: Learning multiple graph centrality measures with a unified network." *International Conference on Artificial Neural Networks*. Cham: Springer International Publishing, 2019.

[3] Zhu, Yanqiao, et al. "Graph contrastive learning with adaptive augmentation." *Proceedings of the web conference 2021*. 2021.

[4] Jin, Wei, et al. "Self-supervised learning on graphs: Deep insights and new direction." *arXiv preprint arXiv:2006.10141* (2020).

[5] Hu, Ziniu, et al. "Pre-training graph neural networks for generic structural feature extraction." *arXiv preprint arXiv:1905.13728* (2019).

**Questions:**

1. Can the authors provide more evidence to support their basic opinion?
2. Can the authors make further discussion on the differences with existing works?
3. The third module aligns the centrality-guided node representation with "the original pre-trained model (line 320)." How is the "original pre-trained model" obtained? Does this mean in CenPre we need to conduct pre-training twice? The overall workflow of the method needs a clearer elucidation.
4. What does "CenPre uses the original node representation" mean in line 484?

---

> ### Author Response · Authors · 2024-11-23
>
> We appreciate the time and effort you've taken to review our work. We apologize for the confusion caused to you, and we regret not receiving a positive review of this paper from you. We hope the following response can clarify your confusion about this paper, and sincerely hope that you can re-examine our work.
>
>
> >**Answer of Weaknesses#1:**\
> >We thank the reviewer for such a valuable comment.\
> The first point we need to clarify is that, as described in the Abstract and Introduction, we argue that existing SSL-based graph pre-training methods ignore the alignment between the representation and the structure of graphs, rather than claiming that existing graph SSL methods neglect the integration of structural information. We focus not only on learning graph structural information, but more importantly, aligning node representations with structural information to enable node representations to absorb graph structural information without losing their original semantic information. **In brief, this paper focuses on integrating and utilizing graph structure information to improve the quality of node representation, thereby enabling the GNN encoder to better model the graph.**\
>  a) As you mentioned, we do agree that GNNs can effectively capture both semantic and structural information simultaneously. Therefore, in the experiments of our paper, GNNs are also used as the encoder in our CenPre framework. As described in Appendix D, we use GCN or GIN as the encoder in our CenPre. GNNs can learn the structural information of a graph, but essentially, they learn the representation of the current node according to the representation of neighboring nodes. Thus, improving the representation of nodes themselves is the key to enhancing the GNN learning graph structure. In this work, we focus on pre-training for graphs, also known as pre-training Graph Neural Networks (GNNs), which aims to pre-train a GNN encoder to produce better representations for graphs. **The motivation for this work comes from the concept of graph centrality that different nodes in the graph have varying importance. That is, different nodes play different roles in learning graph structure due to their varying degrees of importance in the graph. Therefore, based on the concept of centrality, the proposed CenPre pre-trains GNN encoder by integrating node importance into node representations through learning degree information, enabling the GNN encoder to better learn and utilize the structural information of the graph, thereby achieving better results in downstream tasks.**\
>  b) We thank the reviewer for such a valuable comment with a deep insight into this task. Yes, as you mentioned, we do agree it is also common practice to explicitly consider structural information in existing SSL works. In our work, unlike existing methods that mostly learn graph structure information by disturbing or reconstructing graphs, this paper integrates node importance into node representation from both node-level and graph-level perspectives. On this basis, we propose an alignment module to align the pre-trained node representation with the original one to learn graph structure information without losing the original semantic information. The experimental results indicate that our method achieves better performance in downstream tasks compared to existing works.\
> >Thanks a lot for your insightful comment, we have revised the related content in the paper to avoid confusion. Please refer to the words in blue in the Abstract and Introduction of the revised manuscript.
>
>
> >**Answer of Weaknesses#2:**\
> >We thank the reviewer for providing such a thoughtful comment. Unlike the mentioned studies, in our work, we first explore the role of different nodes in learning graph structure by leveraging the importance of nodes in the graph based on degree centrality. Further, we learn the important neighborhood information based on Eigenvector centrality, aiming to aggregate the important neighboring information at the graph level when learning graph representation. **In this way, it is possible to effectively leverage the different roles played by different nodes when learning graph structure.**\
> >In the revision, we have added the discussion of these 5 references in the Related Work section. Please refer to the words in blue in the Related Work Section.
>
>
> >**Answer of Weaknesses#3:**\
> >Thanks for the valuable suggestions. We have shortened the length of Sec 4.1 and Sec 4.2 in the revision by moving some content to the Appendix.

---

> ### Author Response · Authors · 2024-11-23
>
> >**Answer of Questions#1:**\
> >We thank the reviewer for such an insightful question. The motivation for this work comes from the concept of graph centrality. We learn the importance of nodes from node-level and graph-level perspectives based on graph centrality, and make node representations with similar importance levels similar. The GNN encoder can learn graph structure in a better way when modeling graph representation. We have revised the description of our motivation and opinion based on your suggestions in the revision. Please refer to the words in blue in the Abstract and Introduction of the revised manuscript.
>
>
> >**Answer of Questions#2:**\
> >We thank the reviewer for the valuable question. As mentioned above, we have included further discussion on the differences with existing works. Please refer to the words in blue in the Related Work section of the revised manuscript.
>
>
> >**Answer of Questions#3:**\
> >We apologize for the confusion caused to you. Here, the "original pre-trained model" represents any graph pre-training model or a GNN encoder, which corresponds to the encoder used in our CenPre. In our work, we use GNN as an encoder, so we have revised the related content to avoid confusion. Please refer to the words in blue in Sec 4.3.
>
>
> >**Answer of Questions#4:**\
> >Thank you for your question. The meaning of "CenPre uses the original node representation" is that our CenPre uses the original node representation from the dataset for graph pre-training.
>
>
> We sincerely appreciate the reviewer once again for your insightful and valuable comments and suggestions, which are significant in improving the quality of this paper. Based on your comments and suggestions, we have responded accordingly and made corresponding modifications to the manuscript. I hope the above response will allow you to reassess our work. Thank you very much!

---

> ### Author Response · Authors · 2024-11-25
> **Request for Your Feedback on Our Response**
>
> Dear Reviewer eSwp,
>
> We hope this message finds you well.\
> We are writing to kindly remind you that the discussion phase is ending soon.
>
> We have included explanations and discussions in our response and made appropriate modifications to the paper based on your comments and suggestions. We would sincerely appreciate it if you could take valuable time out of your busy schedule to review our response. Your timely feedback will help us confirm whether we have appropriately addressed the issues and suggestions you raised.
>
> We would greatly appreciate any updates or further feedback from you. Your insights and expertise are invaluable to us, as we aim to address any concerns and improve our submission.
>
> Thank you for your time and attention. We look forward to hearing from you soon.
>
> Best regards,\
> Authors of Submission6687

---

> ### Comment · Reviewer_eSwp · 2024-11-25
>
> Thanks for the detailed response from the authors. Some of my concerns have been addressed. However, some key points are still not clear.
>
> The two concepts "the alignment between the representation and the structure" and "the integration of structural information" seem quite intertwined. In fact, in the current manuscript or even in the response, we can see many expressions similar to the latter.
>
> Anyway, in the revised version, the focus of the paper has been shifted to the "varying importance," suggesting that the different roles/importance of nodes should be considered in the pre-training process. This makes sense to me. However, some discussion on related work should be provided. The revised manuscript simply lists these works without elaborating on the key differences/advantages of the proposed CenPre compared with them, especially [4,5], which exactly use centrality as a pretext task for pre-training. How is the proposed CenPre fundamentally different from them? It seems to me CenPre looks similar to them and lacks novelty, which is my primary concern currently.
>
> Based on the above reasons, I will maintain my rating currently, and welcome further explanations from the authors.

---

> > ### Author Response · Authors · 2024-11-27
> > **Thank You for Your Insightful Feedback**
> >
> > Dear Reviewer eSwp,
> >
> > Thank you very much for your detailed feedback and for highlighting both the strengths and remaining concerns of our manuscript.
> >
> >
> > 1. We are pleased to have found common ground during our previous exchange. As you mentioned, based on your comments and suggestions, in the revised version, the focus of the paper has been shifted to "varying importance," suggesting that the different roles/importance of nodes should be considered in the pre-training process. We are deeply grateful for the time and effort you have dedicated to providing thoughtful, constructive, and insightful feedback, which has been invaluable in enhancing our research.
> >
> > 2. We sincerely appreciate the reviewer for providing several references on centrality, especially two remarkable references [4] and [5], which are crucial for us to strengthen the description of the differences between our paper and existing research. We will analyze in detail the fundamental differences between our work and references 4 and 5 to demonstrate the novelty of our work.
> > (a) **Comparison with Referece[4].** Referece[4] only uses node degree for its centrality-based learning, limiting its ability to capture global graph structures. From the results in Table 2 and Table 5 (the green text in tables), we can see that our CenPre performs significantly better than NodeProperty[4], which implies that relying solely on node degree in centrality-based learning will limit the ability to capture global graph structures. Further, the results in Table 5 show that whether using only node-level structure (w/ ${\mathcal{L}}_{d}$) or graph-level structure (w/ ${\mathcal{L}}_{e}$), the performance of the model has significantly decreased. This verifies that we need to simultaneously explore the different importance of nodes from both local and global perspectives in order to better learn the structural information of the graph. We have added the results and corresponding analysis in the revision, please refer to the green text in Sec 2.2 and Sec 5.3.\
> > (b) **Comparison with Referece[5].** Referece[5] uses centrality score ranking to guide GNNs in capturing structural features. However, focusing solely on the centrality score ranking may fail to effectively capture meaningful features. Because a slight modification to a graph, such as the removal of a few edges, may significantly alter centrality values without necessarily impacting the ranking. In this case, we can only learn structural information by learning the actual value of the centrality score. Here, we demonstrate this property using a simple example graph with 7 nodes (as shown in Figure 4 in the revision). Assuming the centrality scores of these 7 points are: A=6, B=2, C=3, D=2, E=2, F=2, G=5. When the edge between node A and node G is removed, the degree of both nodes A and G changes (i.e. the centrality scores of these 7 points have been changed to: A=5, B=2, C=3, D=2, E=2, F=2, G=4). It is obvious that the structure of this graph has changed, and the importance of nodes has also changed, but their rankings based on degree centrality remain unchanged. In this case, Referece[5] is difficult to learn meaningful structural information based on the centrality ranking. This illustrates that relying solely on degree order-based centrality fails to capture structural changes in the graph, as evidenced by the significant updates in eigenvalues following the edge removal. For our method, we use the actual value of Eigenvector centrality to capture the global information of the graph, which effectively identifies the structural differences between the two graphs based on the importance of different Eigenvector centrality values of nodes, thereby meaningful graph structural information can be captured. In addition, In order to enable our method to learn graph structure information without losing semantic information, we also propose a graph representation alignment module. As shown in the results of Figure 6, this module can effectively improve the stability of graph representation. Meanwhile, based on the results of Table 5, it can be seen that the removal of the alignment-based loss results in a significant performance drop. This indicates that our graph representation alignment module effectively improves the graph representation. We have added the results and corresponding analysis in the revision, please refer to the green text in Appendix F.
> >
> > In closing, we once again express our gratitude for your constructive feedback, which has significantly strengthened the quality of our manuscript. Your thoughtful suggestions have not only helped us refine our work but also encouraged us to delve deeper into the nuances of our methodology and its positioning within the field. We hope our clarifications and revisions address your concerns and look forward to any additional feedback you may have.
> >
> > Thank you for your commitment to fostering rigorous and impactful research.
> >
> > Best regards,\
> > Authors of Submission6687

---

> > > ### Comment · Reviewer_eSwp · 2024-11-28
> > >
> > > Thanks for the detailed response. The additional discussion elucidated the differences between CenPre and these prior works. However, I feel that these methods all share the core idea of aligning pretrained representations with centrality, and the uniqueness of CenPre seems somewhat minor and not sufficiently new. Compared to [4], CenPre differs by incorporating eigenvectors, which have been widely used in the GNN domain. Compared to [5], CenPre directly aligns centrality score instead of ranking, but the benefits of doing so are illustrated by some special cases without solid theoretical justification.
> > >
> > > I agree that minor innovation does not necessarily mean minor contribution, as long as it is solid enough and helpful to the field. For this work, I believe it could be more qualified with some theoretical justifications, considering that its core design is not new.
> > >
> > > I will slightly increase my rating to 5, taking into account the good writing and the author's meticulous efforts in engaging in discussions and improving the manuscript. Finally, I have some questions:
> > >
> > > * Returning to the Weakness 3 I initially raised, the current manuscript seems to suggest that $f_o$ is a randomly initialized, untrained GNN encoder. Is this understanding correct?
> > >
> > > * Centrality (including degree and eigenvector) has been widely used as positional/structural encoding in graph transformers. What sets CenPre apart from them?

---

> ### Author Response · Authors · 2024-12-02
> **Response to Reviewer eSwp's Feedback [1/4]**
>
> Firstly, we would like to express our sincere gratitude to the reviewer for improving the rating of this paper. We would greatly appreciate your further replies and comments.\
> Here, we would like to clear that, unlike previous studies that directly predict or rank the node representation based on centrality score, our method proposed a novel graph-level structure learning module to characterize the importance between all nodes in the graph based on eigenvector centrality, enabling the exploitation of graph-level structure similarities/differences when learning node representation. **Our approach is not to directly align centrality scores, but to learn commonalities with similar/same centrality and distinguish between representations with different centrality based on contrastive learning.**\
> As described in Lines 69-80, the idea of our graph-level structure learning module comes from contrastive learning, in which we propose a Contrastive Representation-Structure Pre-Training (CReSP) strategy that aligns node representations with the graph’s structural pattern. Compared with previous studies, our method can improve the robustness and generalizability of graph representation by learning common features of similar centrality and distinguishing different centralities of data based on the merit of contrastive learning. Below, we provide relevant derivation and experimental results to demonstrate our proposed $\mathcal{L}_e$ regarding stability without the alignment.
>
> **a) The stability of $\mathcal{L}_e$ in gradient updates.:**
>
> We first categorize the operations in this framework into three types:
> * $\textbf{T1.}$ the interaction between two trainable vectors (such as the inner product between weight matrix/vectors and node representations).
> * $\textbf{T2.}$ the feed-forward layer with vectorized inputs and outputs.
> * $\textbf{T3.}$ the interaction between predictive vectors and golden labels.
>
> Accordingly, the proof contains three parts:
> * $\textbf{P1.}$ If the two vectors in $\textbf{T1}$ are from two independent Gaussian distributions, then the output of $\textbf{T1}$ follows Gaussian.
> * $\textbf{P2.}$  If the input of $\textbf{T2}$ is Gaussian, then the distribution of $\textbf{T2}$ output is bounded with high probability.
> * $\textbf{P3.}$  If $\textbf{T3}$ is based on the above assumptions of $\textbf{T1}$ and $\textbf{T2}$, then for any two training samples, the difference of their gradients is proximately bounded.

---

> ### Author Response · Authors · 2024-12-02
> **Response to Reviewer eSwp's Feedback [2/4]**
>
> The proof of $\textbf{P1}$ and $\textbf{P2}$ are straightforward and intuitive; therefore, we present only the basic ideas here.
>
> $\textbf{P1}$ Suppose the representation vector of a node $r=\lbrace r_1,r_2,...,r_n \rbrace$ and weight matrix $W=\lbrace w_{ij} \rbrace, i = 1,2,...,n, j=1,2,...,k$ are randomly initialized by $r_i \sim \mathcal{N}(0,\sigma_r^2)$, and  $w_i \sim \mathcal{N}(0,\sigma_w^2)$.
>
> Then according to normal product distribution, each product term of $w_{ij}r_i$, which has a distribution with mean zero and variance proportional to the product of the variances:$\mathbb{E}(w_{ij}r_i) = 0$, $\mathit{Var}(w_{ij}r_i) = \sigma_r^2\sigma_w^2$.
>
> If we suppose the output of $\textbf{T1}$ is $z=Wr$ where $z={z_1,z_2,...,z_k}$ and $z_j = \Sigma_{i=1}^nw_{ij}r_i$, we have $z_j \sim \mathcal{N}(0,n\sigma_r^2\sigma_w^2)$.
>
> Then, we can claim that for any $\epsilon \geq 0$, there is a $\delta \geq 0$, such that $P(||z||>\delta)\leq \epsilon$.
>
> **Note**: we can add a layer normalization operator after $\textbf{T1}$ to further bound the distribution within a smaller variance.
>
> $\textbf{P2:}$ Considering Heine-Cantor theorem on $\textbf{T2}$, where all the mapping functions in $\textbf{T2}$ are continuous on $\mathcal{R}^n$, and input is a bounded Gaussian with high probability, the output should be bounded as well. We describe this property by Lipchitz continuity:
> For any pair of inputs $z_a$ and $z_b$ in $\textbf{T2}$, there is a constant $K$ which bounds the corresponding output $O_a$ and $O_b$ by: $||O_a - O_b||\leq K||z_a - z_b||$
>
> Then, we have $\frac{\partial O_a}{\partial z_a}\leq K$ if we force $z_b$ approximate to $z_a$.
>
> $\textbf{P3:}$  Considering the probabilistic-based converge, the previously discussed progress can be described as:
>
> For any $\epsilon \geq 0$, there is a $\delta \geq 0$, and a content $K \geq 0$, for any pair of inputs $r_a$ and $r_b$, after the operation defined in $\textbf{T1}$ and $\textbf{T2}$, we have $||O_a - O_b|| < K\delta$ is true with probability $1-\epsilon$.
>
> In our training, each node will be updated $n$ times, $1$ as positive sample and $n-1$ as negative samples, and the objective label is based on the index, which means that $||\mathcal{Y}^e_i|| = 1$.
>
> Assume the $\mathcal{P}_e(r_i;\theta_e)$ is formed by $\textbf{T1}$ and $\textbf{T2}$ sequentially, then we simplify the above equation by the notation of $O_a$ and $z_a$ for any input representation $r_a$ as: $\mathcal{Y}^e_a \frac{1}{ O_a}\frac{\partial O_a}{\partial z_a}\frac{\partial z_a}{\partial r_a} \leq \frac{WK}{ O_a}$
>
> Then, in each round of updating, for one specific dimension in $r_a$, noted as $r_{ai}$, the gradient is $\mathcal{Y}^e_a \frac{1}{ O_a}\frac{\partial O_a}{\partial z_a}\frac{\partial z_a}{\partial r_{ai}} \leq \mathcal{Y}^e_a\frac{K}{O_a}\sum_{j=1}^k{w_{ij}}$
>
> **Note:** $\sum_{j=1}^k{w_{ij}}$ follows normal distribution since each $w_{ij}$ was initialized from a normal distribution independently.
>
> For another sample $r_b$, the gradient for the same dimension is: $\mathcal{Y}^e_b \frac{1}{ O_b}\frac{\partial O_b}{\partial z_b}\frac{\partial z_b}{\partial r_{bi}} \leq \mathcal{Y}^e_b\frac{K}{O_b}\sum_{j=1}^k{w_{ij}}$
>
> Considering the $||\mathcal{Y}^e_a||$ and $||\mathcal{Y}^e_b ||$ are all identifiable representations of labels (only one dimension is $1$ and others are $0$), we then have $||\mathcal{Y}^e_a|| = ||\mathcal{Y}^e_b|| = 1$. Besides, the $O_a$ and $O_b$ are predictive outputs that are usually normalized by activation such as softmax.
>
> For a specific dimension, we conclude that the gradients for updating should remain on the same scale. Moreover, each training sample is treated as a positive sample once and as a negative sample $n-1$ times. This implies that the updates to the representations are consistently scaled for each training sample and each identifiable label. In contrast, the learned representations may become biased when predicting non-identifiable labels or dealing with imbalanced category types. As highlighted in Ref[5], directly optimizing the eigenvalues can yield negative outcomes, as it may be initially caused by the imbalanced distribution in $\mathcal{Y}^e_b$ and subsequently in $O_i$.
>
> **Note:** The above derivation demonstrates that the proposed $\mathcal{L}_e$ is capable of learning stable representations with high probability, even in the absence of a regularization term $\mathcal{L}_r$. Moreover, the inclusion of $\mathcal{L}_r$ in our framework further enhances stability, as evidenced by the ablation studies presented in Tables 5 and 6.

---

> ### Author Response · Authors · 2024-12-02
> **Response to Reviewer eSwp's Feedback [3/4]**
>
> **b) The comparison results of our alignment idea:**
>
> To produce more robust and generalizable graph representations, we propose a Contrastive Representation-Structure Pre-Training (CReSP) strategy that aligns node representations with the graph’s structural pattern, enabling the exploitation of graph-level structure similarities/differences when learning node representation. To demonstrate the effectiveness of our method in downstream tasks, we compare our CenPre with Ref[5], directly predict Eigenvector Centrality scores, directly align centrality scores without contrastive learning, and report the results in Table 1.
>
> *Table 1. Micro F1 (\%) results of directly predicting Eigenvector Centrality scores (Predict Eigen Scores Only), directly align centrality scores with prediction rather than using contrastive learning (Replace $\mathcal{L}_e$ with Eigen Scores Prediction in our CenPre ), Ref[5], and our CenPre.*
>
> | Methods | Cora(Node) | Pubmed(Node) | IMDB-B(Graph) |
> | :---: | :---: | :---: | :---: |
> | Predict Eigen Scores Only | 75.9 $\pm$ 1.1 | 73.5 $\pm$ 2.4 | 72.5 $\pm$ 0.9 |
> | Ref[5] | 81.9 $\pm$ 1.0 | 79.1 $\pm$ 0.9| 73.2 $\pm$ 0.8 |
> | Replace $\mathcal{L}_e$ with Eigen Scores Prediction in our CenPre | 82.0 $\pm$ 0.5 | 79.6 $\pm$ 1.2 | 74.8 $\pm$ 1.5 |
> | **CenPre (ours)** | **83.9 $\pm$ 0.3** | **81.9 $\pm$ 0.5** | **77.9 $\pm$ 0.9** |
>
> From the results in Table 1, we can see that our CenPre performs consistently better than the baseline models, indicating the effectiveness of our method in graph pre-training. Compared with the results of "Predict Eigen Scores Only" which directly predicts Eigenvector Centrality scores, Ref[5] uses ranking to learn Eigenvector Centrality scores and achieves better performance on all datasets. This implies that the method of learning the relative relationship between two nodes through ranking is more effective in learning graph representations than directly predicting Eigenvector Centrality scores. In addition, compared with the results of "Predict Eigen Scores Only", the results of "Replace $\mathcal{L}_e$ with Eigen Scores Prediction in our CenPre " are better on all datasets. This demonstrates that even predicting Eigenvector Centrality scores can achieve improved performance in our CenPre framework than only predicting Eigenvector Centrality scores ("Predict Eigen Scores Only"), which proves the generalization ability and effectiveness of our method in learning graph representations from another perspective. Further, compared with the results of "Replace $\mathcal{L}_e$ with Eigen Scores Prediction in our CenPre", our CenPre performs significantly better on all datasets. This indicates that using contrastive learning in aligning graph representation based on Eigenvector Centrality can improve the robustness and generalizability of graph representation by learning common features of similar centrality and distinguishing different centralities of data, thereby achieving better performance in downstream tasks.
>
> In summary, the above content demonstrates the effectiveness of the proposed contrastive learning-based centrality alignment method in graph pre-training through theoretical and experimental results.
> We will add the corresponding content in the final version.
>
> [5] Hu, Ziniu, et al. "Pre-training graph neural networks for generic structural feature extraction." arXiv preprint arXiv:1905.13728 (2019).

---

> ### Author Response · Authors · 2024-12-02
> **Response to Reviewer eSwp's Feedback [4/4]**
>
> > 1. Yes, $f_o$ is an untrained GNN encoder, meaning it has not been pre-trained by our CenPre. The purpose of $f_o$ here is to model the original semantic information of nodes so that structural information can be aligned without losing the semantic information of nodes. **The prerequisite reason why untrained GNNs can learn the original semantic information of nodes here is that the original node representations of the dataset we are using are trained through some methods that can capture the semantic information or original properties, such as Word Embeddings, Chemical Properties, etc.** Therefore,  our graph representation alignment module can learn graph structure information without losing semantic information based on the alignment operation. Our work is to improve these representations by pre-training GNNs, aiming to achieve better performance in downstream tasks. Our experimental results demonstrate the effectiveness of our method.
>
> > 2. As mentioned above, to produce more robust and generalizable graph representations, different from existing methods, we propose a Contrastive Representation-Structure Pre-Training (CReSP) strategy that aligns node representations with the graph’s structural pattern, enabling the exploitation of graph-level structure similarities/differences when learning node representation. To demonstrate the effectiveness of our method in downstream tasks, we have provided theoretical and experimental evidence to demonstrate the rationality and effectiveness of our method. We will add the corresponding theoretical proof and experimental results in the final version.
>
>
> Finally, we would like to express our heartfelt gratitude to the reviewer for the constructive comments. Based on your comments and suggestions, we will make corresponding revisions in the final version to improve the quality of the paper. We sincerely hope you can re-evaluate our work. Thank you very much for your effort and time in reviewing this paper.

---

> ### Comment · Reviewer_eSwp · 2024-12-03
>
> Thanks for the detailed response. My concerns have been addressed. I will adjust my rating accordingly. As a side note, the phrase "structure learning" is used many times in the manuscript, which may be confused with another line of work [1]. It is recommended to replace it with a different expression.
>
> [1] Zhu, Yanqiao, et al. "A Survey on Graph Structure Learning: Progress and Opportunities."

---

> > ### Author Response · Authors · 2024-12-03
> >
> > Dear Reviewer eSwp,
> >
> > We would like to express our sincere gratitude to you for improving the rating of this paper and providing further insightful feedback. We will make corresponding modifications in the final version based on your valuable suggestions. Thank you once again for the time and effort you have put into reviewing this paper. Thank you very much!
> >
> > Best regards,\
> > Authors of Submission6687

---

### Official Review · Reviewer_7DHX · 2024-10-30

**Soundness:** 3
**Presentation:** 3
**Contribution:** 2
**Rating:** 6
**Confidence:** 5

**Summary:**

Summary:
The author proposed CenPre framework enhances graph representations by integrating structural information, specifically node importance via centrality measures, into self-supervised learning. CenPre includes modules for node-level structure learning using degree centrality, graph-level structure learning with eigenvector centrality, and a representation alignment module that retains semantic integrity while incorporating structural insights. Experiments show that CenPre outperforms current methods in node classification, link prediction, and graph classification.

**Strengths:**

- The author proposed a new method CenPre framework enhances graph representations by integrating structural information, specifically node importance via centrality measures, into self-supervised learning.
- Experiments show that CenPre outperforms current methods in node classification, link prediction, and graph classification.

**Weaknesses:**

- More self-supervised learning methods with different training paradigms are encouraged to be added. GBT[1] using barlow twins and GGD[2] using group discrimination.
- There are three hyperparameters λ1/2/3 to be tuned. It seems it is quite hard to tune these parameters.
- Has the pretraining methods tried to employed to be pretrained on a set of other datasets first and then evaluate on a target dataset? It seems the method is still dataset-specific? The L_d and L_e only relies on graph topology and seems to be able to used in this setting.


[1] Bielak, P., Kajdanowicz, T., & Chawla, N. V. (2022). Graph barlow twins: A self-supervised representation learning framework for graphs. Knowledge-Based Systems, 256, 109631.
[2] Zheng, Y., Pan, S., Lee, V., Zheng, Y., & Yu, P. S. (2022). Rethinking and scaling up graph contrastive learning: An extremely efficient approach with group discrimination. Advances in Neural Information Processing Systems, 35, 10809-10820.

**Questions:**

same as cons

---

> ### Author Response · Authors · 2024-11-23
>
> We appreciate the time and effort you've taken to review our work.
>
> > **Weaknesses#1:**\
> > More self-supervised learning methods with different training paradigms are encouraged to be added. GBT[1] using barlow twins and GGD[2] using group discrimination.
>
> **Answer of Weaknesses#1:**\
> We thank the reviewer for such a valuable comment. We have added the discussion of the mentioned references in the revision. Please refer to the Related Work section in the revision. In the final version, we will try to conduct experiments and comparisons based on the mentioned references.
>
>
> > **Weaknesses#2:**\
> > There are three hyperparameters λ1/2/3 to be tuned. It seems it is quite hard to tune these parameters.
>
> **Answer of Weaknesses#2:**\
> We thank the reviewer for such an insightful comment. For model parameters, we use grid search to find the optimal parameters. For the three hyperparameters λ1/2/3, in preliminary experiments, we found that the performance of the model is stable within a certain range of values for these three hyperparameters. Therefore, for the scalability of our method, we set $\lambda_1 = 1$, $\lambda_2 = 1$, and $\lambda_3=5$ to make them to the same scale. Moreover, the best performance was achieved by this setting in all datasets of the three tasks. Thanks again for your comment, we have included the corresponding explanation in the revision. Please refer to the words in blue in Appendix D.
>
>
> > **Weaknesses#3:**\
> > Has the pretraining methods tried to employed to be pretrained on a set of other datasets first and then evaluate on a target dataset? It seems the method is still dataset-specific? The L_d and L_e only relies on graph topology and seems to be able to used in this setting.
>
> **Answer of Weaknesses#3:**\
> Thank you very much for the valuable comment. to evaluate the transferability of the pre-training scheme, we conduct experiments in the transfer learning scenario, which pre-trains our method on a dataset first and then evaluates the model on a different target dataset. The following table shows some experimental results of transfer learning in graph classification, where we pre-train our proposed CenPre on the ZINC-2M dataset, and evaluate our CenPre on BBBP, Tox21, ToxCast, and SIDER datasets.  From the results, we can see that our CenPre consistently performs better than the baseline models, which demonstrates the effectiveness of our CenPre in transfer learning. Due to time constraints, we are unable to run all the datasets during the rebuttal phase. We will continue to run other datasets and include complete experiments and corresponding discussion in the final version.
>
> *Table 1. Experimental results in transfer learning. (\%)*
>
> | Model | BBBP | Tox21 | ToxCast | SIDER |
> |:----:|:----:|:-----:|:----:|:----:|
> |No-pretrain |65.80 | 74.00 |63.40 | 57.30 |
> |GraphCL| 69.68| 73.87 | 62.40 | 60.53 |
> |JOAO| 70.22 | 74.98 | 62.94 | 59.97 |
> |SimGRACE| 71.25 | 75.21 | 63.36 | 60.59 |
> |CenPre (ours)| **73.15** | **76.89** | **65.12** | **61.93** |

---

### Official Review · Reviewer_rWJN · 2024-11-01

**Soundness:** 2
**Presentation:** 2
**Contribution:** 2
**Rating:** 6
**Confidence:** 4

**Summary:**

In this paper, the authors propose the CenPre framework, which aligns graph structure information with representations based on the concept of centrality. By learning importance at both the node and graph levels, CenPre enhances node representation quality through pretraining and alignment modules.

The main contributions include:

Aligning graph representations with structural information during pretraining to generate improved graph representations for downstream tasks.
Introducing the CenPre framework based on centrality, capable of learning structure-integrated graph representations from both local and global perspectives.
Demonstrating, through experiments on multiple real-world datasets, that CenPre significantly outperforms baseline models in node classification, link prediction, and graph classification tasks.

**Strengths:**

1、The authors propose a novel centrality-guided graph pretraining framework, CenPre, which integrates graph structural information with node representations. This systematic application of the centrality concept in graph pretraining is innovative.

2、Unlike traditional graph representation learning methods that rely solely on augmentation, this paper introduces an innovative approach by learning node importance at both the node and graph levels, with corresponding modules designed for structural information integration.

3、The experimental section conducts a comprehensive evaluation on 13 benchmark datasets across various domains and scales, covering node classification, link prediction, and graph classification tasks. The rigorous experimental setup and comparisons with numerous advanced baseline models enhance the reliability and persuasiveness of the results.

**Weaknesses:**

1、The choice of centrality metrics by the authors is relatively traditional, lacking exploration of alternative, potentially more suitable centrality measures for complex graph structures or specific application scenarios.

2、There is an absence of sensitivity analysis on the model under different parameter settings, making it difficult for readers to understand the stability and performance trends of the model with various hyperparameter values.

3、The theoretical explanation of the relationship between centrality, graph structure, and node representation lacks depth, missing a more rigorous mathematical basis to explain why this method can effectively improve graph representation learning performance.

4、The analysis of computational and spatial complexity is not comprehensive, with a lack of detailed efficiency comparisons to existing methods.

**Questions:**

1、The paper employs degree centrality and eigenvector centrality to measure node importance. What are the specific reasons for choosing these two centrality metrics? Were other centrality measures considered, and if so, why were they not selected? It is recommended to explore a broader range of centrality metrics and analyze their suitability for different types of graph data and tasks. Experimental comparisons of the CenPre framework's performance under different centrality metrics could further optimize how node importance is measured.

2、The CenPre framework consists of three modules. How do these modules interact with each other? For instance, when integrating information between the node-level and graph-level structural learning modules, is there interference or synergistic enhancement? How could experimental or theoretical analysis be used to understand the interactions between these modules more deeply? A more detailed experimental design, examining the impact of various module combinations on model performance, would allow for an in-depth analysis of module synergy. From a theoretical perspective, a mathematical model describing the information flow and interactions between modules could provide a foundation for further optimizing the framework structure.

3、In the experimental section, for hyperparameter settings such as λ1, λ2, and λ3 in the balanced loss component, it is mentioned that these values were determined through pilot studies, but the specific tuning process and rationale were not elaborated. Could you provide more details on the hyperparameter selection process to help readers better understand the model’s sensitivity and stability?

---

> ### Author Response · Authors · 2024-11-23
>
> We appreciate the time and effort you've taken to review our work.
>
> >**Answer of Weaknesses#1:**\
> >We thank the reviewer for such an insightful comment. I completely agree with your viewpoint, this is one of the limitations of our method. As described in the Limitations section of Appendix F, the performance of our method may be sensitive to the choice of centrality measures, requiring careful tuning and selection based on specific graph characteristics. However, in this paper, to make our method more generalizable, we design a simple method of using centrality metrics. The experimental results demonstrate the effectiveness of our method. Thanks again for your comment, we have included the corresponding content in the revision. Please refer to the words in blue in the Limitations section of Appendix F in the revision.
>
> >**Answer of Weaknesses#2:**\
> >We thank the reviewer for such an insightful comment. In our experiments, we use grid search to find the optimal parameters. Further, the experimental results of our models are averaged over 10 runs with different random seeds to ensure the final reported results are statistically stable. Thanks again for your comment, in the revision, we have conducted significance tests of our CenPre over the baseline models, and the results show that our CenPre significantly outperforms the baseline models in terms of most of the evaluation metrics (with $p\rm{-}value<0.05$). Please refer to Table 2, Table 3, and Table 4 in the revision. We will add the detailed significance tests and the sensitivity analysis of our CenPre in the final version.
>
> >**Answer of Weaknesses#3:**\
> >We thank the reviewer for the valuable comment. Adding theoretical analysis can indeed make the motivation of this article more solid. Based on your comment, we are deriving mathematical proof from the perspective of the impact of centrality on the distribution of graph representation, and we will include the relevant proof process in the final version.
>
> >**Answer of Weaknesses#4:**\
> >We thank the reviewer for the valuable comment. We have included the Complexity Analysis in the revision. Please refer to Appendix E in the revision.
>
> >**Answer of Questions#1:**\
> >We thank the reviewer for such a thoughtful question. The reasons for using degree centrality and eigenvector centrality to measure node importance in this paper are as follows. Firstly, aiming to explore the role of different nodes in learning graph structure by leveraging the importance of nodes in the graph, we chose degree centrality. In addition, since we used GNN as the encoder, which updates the current node by aggregating neighboring node information, we also used Eigenvector centrality to learn the important neighborhood information of the current node. The ablation study shows the experimental comparisons of the CenPre framework's performance under different centrality metrics, i.e. "w/o $L_d$" and "w/o $L_e$". Please refer to the results in Table 3. Thanks again for your question, in future work, we will further explore other centrality metrics to further improve our CenPre.
>
> >**Answer of Questions#2:**\
> >Thank you very much for this valuable question. We conducted an ablation analysis on our framework in Table 3. From the experimental results, it can be seen that the three modules of our framework synergistically enhance the performance. Removing one of the modules will result in performance degradation of the framework. Thank you again for your question. We are further analyzing the interaction between the three modules and their impact on node representation distribution from different perspectives. Due to time constraints, we will include these experiments and corresponding analyses in the final version.
>
> >**Answer of Questions#3:**\
> >We thank the reviewer for such an insightful question. For model parameters such as λ1, λ2, and λ3, we use grid search to find the optimal parameter combination. Thanks again for your question, we have included the description in the revision. Please refer to the words in blue in Appendix D.

---

> ### Comment · Reviewer_rWJN · 2024-11-26
>
> Thank you for the author's response. My related concerns have been partially addressed.

---

> ### Author Response · Authors · 2024-11-27
> **Thank You for Your Positive Comment and Further Feedback**
>
> Dear Reviewer rWJN,
>
> We sincerely appreciate your valuable comments. This is of great help in improving the quality of our paper.
>
> The following are further supplemental answers to the valuable questions you raised.
>
> **1. Supplement Answer of Questions#1 and Questions#2:**\
> Thanks again for such valuable questions. In the revision, we have included additional experimental results of only using degree centrality or eigenvector centrality. From the experimental results in the following table, we can see that whether using only degree centrality (w/ ${\mathcal{L}}_{d}$) or eigenvector centrality (w/ ${\mathcal{L}}_{e}$), the performance of the model has significantly decreased. This verifies that we need to simultaneously explore the different importance of nodes from both local and global perspectives in order to better learn the structural information of the graph. This also demonstrates the effectiveness of our CenPre in learning graph representations by considering both degree centrality and eigenvector centrality from the perspective of experimental results. Please refer to the green text in Sec. 5.3.\
>
> *Table 1. Experimental results of only using degree centrality (w/ ${\mathcal{L}}_{d}$) or eigenvector centrality (w/ ${\mathcal{L}}_{e}$). (\%). $\Delta$ denotes the performance drop relative to the full CenPre model.*
>
> | Model | Cora-Node | $\Delta$ | Cora-Link | $\Delta$ |  MUTAG-Graph | $\Delta$ |
> |:----:|:----:|:-----:|:-----:|:-----:|:-----:|:-----:|
> |CenPre (ours)| 85.15 | 0.00 | 95.05 | 0.00 | 94.74| 0.00 |
> |w/ ${\mathcal{L}}_{d}$| 81.43 | 3.72 | 92.80 | 2.25 | 91.62 | 3.12 |
> |w/ ${\mathcal{L}}_{e}$| 79.91 | 5.24 | 93.17 | 1.88 | 90.38 | 4.36 |
>
>
> In addition, to analyze the roles of the three modules in our method from a theoretical perspective, we introduce the concept of Scaling Sensitivity vs. Ranking Sensitivity to demonstrate that the three modules in our method are synergistic enhancements. Please refer to Appendix F for the detailed analysis.
>
>
>
> **2. Supplement Answer of Questions#3:**\
> We sincerely appreciate the reviewer once again for the insightful question. In this revision, we have added the experimental results of the sensitivity of the CenPre's performance to the variation of hyperparameters $\lambda_1$, $\lambda_2$, and $\lambda_3$. Please refer to Figure 7 in the revision. The results show that the performance of the model will only fluctuate within a certain range when the values of hyperparameters are set within a reasonable range. For example, when the value range of $\lambda_1$ is 1-4, the fluctuation amplitude of model performance is within 1\%. Further, we can see that both excessively large and excessively small values can lead to a clear decrease in performance. In addition, analysis reveals that the model is most sensitive to $\lambda_3$, showcasing the critical role of the alignment loss in optimizing performance. Based on the results of the three hyperparameters, we can conclude that the model performs best when $\lambda_1=1$, $\lambda_2=1$, and $\lambda_3=5$. One possible reason is this setting makes the three hyperparameters on the same scale, which can lead to better learning of losses. Therefore, we set $\lambda_1=1$, $\lambda_2=1$, and $\lambda_3=5$ in our experiments.\
> We have added the results and corresponding analysis in the revision, please refer to Appendix G.
>
> Please allow us to express our sincere gratitude once again for your valuable comments and suggestions. We will continue to address the remaining issues in the coming time, striving to solve all the problems you raised one by one and present them in the final version. Your insightful feedback is invaluable in improving the quality of this article. Thank you very much for your valuable time and effort.
>
> Best regards,\
> Authors of Submission6687

---

### Official Review · Reviewer_agZV · 2024-11-04

**Soundness:** 3
**Presentation:** 3
**Contribution:** 3
**Rating:** 8
**Confidence:** 3

**Summary:**

The paper introduces a novel self-supervised framework for graphs, which considers aligning graphs' representation and structure based on centrality in graph theory. It contains three modules of node representation pre-training and alignment. The node-level structure learning aims to enhance the node representations based on degree centrality to assign similar embeddings for nodes with the same degree. The graph-level structure learning is based on eigenvector centrality to provide information on node importance from a global point of view. Specifically, it uses the graph representation to guide the structure matrix in determining node importance through a cross-attention module. at last, a graph representation alignment module is applied to minimize the discrepancy between the pre-trained structural embedding and embeddings learned by other graph models. The experimental results demonstrate the effectiveness of the framework.

**Strengths:**

1. It proposes a novel self-supervised framework for pre-training the graph that considers both structure embeddings and graph embedding based on node features only.
2. it introduces a novel alignment module with a loss function that aligns the structure embeddings from a global perspective and node embeddings based on node features only.
3. the experimental results outperform baselines significantly.

**Weaknesses:**

1. The use of SVD limits the applicable ability from very large graphs.
2. based on Table 4, the proposed framework is most effective for sparse graphs. For relatively dense graphs like Photo, the improvement is limited.
3. combined with Tables 4, 5, and 6. the proposed framework is most effective for graph classification while it has limited improvement for node classification and edge classification.
4. some writing errors, e.g., missing "is" in "which the maximum absolute" at line 267, capital "D"at line 245

**Questions:**

1. Why do you choose $L_2$ as your alignment loss function not a distribution alignment loss function like KL-divergence?

---

> ### Author Response · Authors · 2024-11-23
>
> We appreciate the time and effort you've taken to review our work.
>
> > **Weaknesses#1:**\
> > The use of SVD limits the applicable ability from very large graphs.
>
> **Answer of Weaknesses#1:**\
> We thank the reviewer for such a valuable comment and deep insight into our work. We agree that the use of SVD limits the applicable ability to use very large graphs. As described in the Limitations section of Appendix F, we have discussed this limitation that the computational complexity of calculating centrality measures, especially for large-scale graphs, can be high, impacting scalability and efficiency. In fact, this is also one of the most concerning issues for us when carrying out this work. Therefore, to alleviate this limitation, as shown in Eq 8, we used TruncatedSVD in this work to reduce the computational complexity of SVD, allowing our CenPre to be applied to large-scale graphs. From the dataset used, it can be seen that our method can be applied to large-scale graphs such as arXiv and Collab, where the node number and edge number of the arXiv dataset are 169,343 and 2,315,598, and the node number and edge number of the Collab dataset are 235,868 and 1,285,465. They are large-scale graphs and the graphs with the largest number of nodes and edges in the commonly used datasets of this domain. This indicates that our method can be applied to existing commonly used graph datasets and can also be well generalized to other large-scale graph datasets.
>
>
> > **Weaknesses#2:**\
> > based on Table 4, the proposed framework is most effective for sparse graphs. For relatively dense graphs like Photo, the improvement is limited.
>
> **Answer of Weaknesses#2:**\
> We thank the reviewer for such a valuable comment with a deep insight into this work. Yes, as described in the Limitations section of Appendix F, our CenPre's reliance on centrality measures may not fully capture complex, multi-faceted structural properties of graphs, thus our CnePre is limited in dense graphs like Photo. One potential reason may be that we incorporate graph structure information into the graph representation based on centrality, and for dense graphs, the discrimination of the degree of node is limited. Therefore, many nodes may have high degrees in a dense graph, and the advantages of centrality may not be fully utilized, thus limiting the improvement of performance. Thanks for your valuable comment, we have included the above content in the limitation section, and will try to explore an effective method to exploit the node importance in a more differentiated way in the future.
>
>
> > **Weaknesses#3:**\
> > combined with Tables 4, 5, and 6. the proposed framework is most effective for graph classification while it has limited improvement for node classification and edge classification.
>
> **Answer of Weaknesses#3:**\
> We thank the reviewer for such an insightful comment. This is a very interesting discovery. We speculate that one possible reason is that our method incorporates structural information into node representations from both node-level and graph-level perspectives, allowing for better learning and modeling of the entire graph's structural information, thus achieving better performance in graph classification. Thanks again for your thoughtful comments. We will continue to analyze this phenomenon through experiments and further improve the performance of node classification and link prediction in future work.
>
>
> > **Weaknesses#4:**\
> > some writing errors, e.g., missing "is" in "which the maximum absolute" at line 267, capital "D"at line 245
>
> **Answer of Weaknesses#4:**\
> We thank the reviewer for the valuable comment. We apologize for the writing errors. We have corrected them in the revision, and avoid these mistakes in the future.
>
>
> > **Questions#1:**\
> > Why do you choose $L_2$ as your alignment loss function not a distribution alignment loss function like KL-divergence?
>
> **Answer of Questions#1:**\
> We thank the reviewer for this thoughtful question. In the preliminary experiments, we also tried other loss functions, such as KL-divergence, $L_1$, Cosine Distance, etc. We found that the performance of $L_2$ was slightly more stable, so we use $L_2$ in our method. Thank you again for your question. We have added corresponding explanations in the revised manuscript. Please refer to footnote 3 on page 6.

---

### Meta-Review · Area_Chair_3y6o · 2024-12-17

**Metareview:**

This paper proposes a new SSL approach for graph learning, which aims to enhance graph representations by incorporating structural information, and specifically node importance as determined by centrality measures.  The approach is comprised of multiple modules spanning node-level and graph-level structure learning through different centrality types (degree and eigenvector), as well as an alignment phase. The authors compare their work to prior baselines in graph SSL, including some from 2023 and outperform in various tasks including node clasisifcation, link prediction and graph classification.

Reviewers leaned positively on this work, hence the accept recommendation. Yet, there were a few themes of comments that were raised which I encourage the authors to fold into the revision:

- Concerns around time and space complexity (rWJN, agZV)

- Several missing references on notable works in the graph SSL community, e.g. GBT and GGD (as 7DHX mentions).

- Some motivation around the use of centrality measures and why exisitng methods cannot capture this information well could be improved (eSwp, rWJN)

Besides this feedback, the line of work referenced on multi-task graph SSL should be improved with [1,2], and there should be some discussion around non-contrastive methods for graph SSL as well [3].  Nonetheless, the authors' proposal appears complementary in spirit to most of these works and the use of centrality-oriented pre-training independently seems of interest to the community.

[1] Multi-task Self-supervised Graph Neural Networks Enable Stronger Task Generalization (Ju et al, ICLR 2023)

[2] Automated Self-Supervised Learning for Graphs (Jin et al, ICLR 2022)

[3] Link Prediction with Non-Contrastive Learning (Shiao et al, ICLR 2023)

**Additional Comments On Reviewer Discussion:**

Authors addressed several concerns in the rebuttal, overlapping with those raised by reviewers above above.  The smallest among these are clarifications, e.g. around the use of SVD, the effectiveness of the approach on sparse graphs compared to dense graphs, and graph classification compared to other tasks in response to agZV.  The authors also added complexity analysis as well as significance tests for their method in response to rWJN.  The authors also included some transfer learning experiments in response to 7DHX, and some proof sketches around the stability of gradient updates in response to eSwp.

---

### Decision · Program_Chairs · 2025-01-22

Accept (Poster)